# Risk-Averse and Optimistic Advertiser Incentive Compatibility in Auto-bidding

**Christopher Liaw** [1]   **Wennan Zhu** [1]

## Abstract

The rise of auto-bidding in online advertising has created new challenges for ensuring advertiser incentive compatibility, particularly when advertisers delegate bidding to agents with high-level constraints. One challenge is the multiplicity of equilibria with reported constraints. Alimohammadi et al. (2023) proposed a notion of Auto-bidding Incentive Compatibility (AIC) which serves to highlight that standard auctions may not incentivize truthful reporting of these constraints. However, their definition of AIC is very stringent as it requires that the worst-case outcome of an advertiser's truthful report is at least as good as the best-case outcome of any of the advertiser's possible deviations. In this paper, we introduce two refined and relaxed concepts: Risk-Averse Auto-bidding Incentive Compatibility (RAIC) and Optimistic Auto-bidding Incentive Compatibility (OAIC). RAIC (OAIC) stipulates that truthful reporting is preferred if its least (most) favorable equilibrium outcome is no worse than the least (most) favorable equilibrium outcome from any misreport. We demonstrate that SPA satisfies both RAIC and OAIC. These findings clarify SPA's incentive properties under auto-bidding, specifically regarding advertiser perspectives on equilibrium selection.

## 1. Introduction

Online advertising increasingly relies on auto-bidding, a method that empowers advertisers to define high-level objectives and constraints, thereby avoiding the manual bid adjustments for each keyword. For example, an advertiser might aim for maximal conversion volume, contingent upon meeting a target cost per acquisition (tCPA) requirement

---

[1]Google Research. Correspondence to: Christopher Liaw <cvliaw@google.com>.

*Proceedings of the $43^{rd}$ International Conference on Machine Learning*, Seoul, South Korea. PMLR 306, 2026. Copyright 2026 by the author(s).

and staying within a specified budget. Other objectives, like maximizing the quantity of clicks, or alternative constraints, such as return-on-spend (ROS), are also common. Auto-bidding agents are designed to solve these optimization problems for advertisers, which not only saves considerable time and effort but can also yield superior advertising outcomes.

From the auto-bidders' perspective, the primary objective is to identify optimal bidding strategies that maximize a specific goal while adhering to advertiser constraints (Aggarwal et al., 2019; Balseiro et al., 2015). From the auction design standpoint, research can be categorized into several areas: the existence and computational complexity of equilibria (Aggarwal et al., 2019; Balseiro et al., 2015; Conitzer et al., 2022; Li & Tang, 2024), allocation efficiency (Aggarwal et al., 2019; Deng et al., 2021; Mehta, 2022; Liaw et al., 2023; 2024), and auction design to maximize revenue (Golrezaei et al., 2021; Balseiro et al., 2021b; 2024). A comprehensive overview of auto-bidding in online advertising can be found in the survey by Aggarwal et al. (Aggarwal et al., 2024a).

Recent studies (Alimohammadi et al., 2023; Feng et al., 2024; Li & Tang, 2024; Wang et al., 2024) have begun to explore the incentives for advertisers to truthfully report their constraints to auto-bidders. Alimohammadi et al. (Alimohammadi et al., 2023) model the auto-bidding problem as a two-stage game. In the first stage, advertisers communicate their constraints to the auto-bidders. Subsequently, the auto-bidders place bids and achieve a subgame equilibrium. They introduce the concept of auto-bidding incentive compatibility (AIC), defining a mechanism as AIC if any auto-bidding equilibrium resulting from advertisers reporting their actual constraints is at least as favorable as any equilibrium achieved when they report different constraints. Under this stringent definition, they show that both First-Price Auction (FPA) and Second-Price Auction (SPA) fail to be AIC in scenarios involving both tCPA and budget constraints.

We aim to establish a method for defining an advertiser's ordinal preferences across various reporting constraints and to assert incentive compatibility, which we define as: a mechanism where an advertiser never strictly prefers reporting constraints other than their actual constraints. While the

AIC definition presented in Alimohammadi et al. (2023) effectively highlights important vulnerabilities in standard auctions, extending it directly to ordinal preferences is challenging because transitivity may not hold when multiple equilibria exist.[1]

## 1.1. Our Contributions

This paper introduces two new concepts of auto-bidding incentive compatibility. First, we say a mechanism is risk-averse autobidding incentive compatible (RAIC) if the least favorable outcome under truthful reporting of their constraints is at least as good as the least favorable outcome under any misreporting of their constraints. We similarly say a mechanism is optimistic auto-bidding incentive compatible (OAIC) if the most favorable outcome under truthful reporting is at least as good as the most favorable outcome under any misreport.

The difference between our definition of RAIC and OAIC with the original definition of AIC proposed in (Alimohammadi et al., 2023) is on equilibrium selection from the set of all possible equilibria with reported targets. Consider the setting with tCPA constraints for example. An advertiser is AIC if the worst equilibrium when they report their true target is not worse than the best equilibrium when they report a different target. For RAIC and OAIC, either the best or the worst equilibrium is chosen for both the advertiser's true target or an alternative reported target. Thus, we can define advertiser's ordinal preferences over different reporting targets.

Our main results are summarized in Table 1. We study tCPA advertiser incentive compatibility under a second-price auction where there are a set of queries and a single winner for each query. Our first result in Section 4 assumes that bidders may place completely arbitrary and unrelated bids on every query and show that SPA is both RAIC and OAIC. However, it is known that uniform bidding, where a bidder places a fixed multiple of their value in every query, is an optimal bidding function. This further constrains the bidder's actions and makes it more technically challenging to understand incentive compatibility. With the assumption that the advertisers bid uniformly, we show that SPA for two advertisers is both RAIC and OAIC. For the first-price auction, it turns out to be a fairly straightforward observation that if uniform bidding is enforced then FPA is AIC (Al-

---

imohammadi et al., 2023), which implies that FPA is both OAIC and RAIC. In Appendix A.3, we also prove that FPA is OAIC with non-uniform bidding.

[1]Suppose we used AIC as a notion of preference. If reports $T$ and $T'$ yield identical sets of equilibria with advertiser values $W_1 > W_2$, AIC compares the worst-case truthful outcome to the best-case deviation. Thus, comparing the worst-case of $T$ ($W_2$) against the best-case of $T'$ ($W_1$) would imply a strict preference $T' \succ T$. Symmetrically, comparing the worst-case of $T'$ ($W_2$) against the best-case of $T$ ($W_1$) would imply $T \succ T'$. This mutual strict preference indicates that transitivity is not maintained under this specific preference model.

**Our techniques.** In non-uniform bidding scenarios, we demonstrate that SPA are both RAIC and OAIC by constructing bids that form an equilibrium when one advertiser reports a lower target. The construction ensures the misreporting advertiser wins only a subset of queries, all bidders meet their tCPA constraints, and no bidder can win additional queries without violating these constraints.

The setting where we assume that bidders bid uniformly is much more technically challenging. Here, we study incentive compatibility when there are two advertisers. As a first step, we establish a characterization of all possible equilibria. In particular, we begin by observing that if we sort the queries in decreasing order of the ratios of the values between advertiser 1 and 2 then every equilibrium always assigns a prefix of the queries to bidder 1 and the remainder to bidder 2 (see Eq. (1) for details). This allows us to establish a total ordering on the possible equilibria $N_0 \prec N_1 \prec N_2 \prec \dots$. This ordering corresponds to the advertiser's preference of outcomes (so, in particular, $N_0$ is the least preferred outcome for advertiser 1, which we can think of as an equilibrium where advertiser 1 wins nothing). Note that not all such equilibria may exist.

Next, we derive exact conditions that are equivalent to the existence of the equilibrium $N_k$ for the reports provided by the advertisers. Given auto-bidders' uniform bid multipliers, it is straightforward to verify if the multipliers form an equilibrium $N_k$. However, it is non-trivial to calculate the set of feasible bid multipliers that forms an equilibrium $N_k$. In Theorem 4, we show an equilibrium $N_k$ exists if and only if a set of inequalities are satisfied. Note that these inequalities only depend on advertisers' reported targets and their value for the queries. Thus, we simplify the task of finding bid multipliers that form an equilibrium to checking a set of inequalities.

We now proceed as follows. First we assume that advertiser 1 specifies a report $T_1$ while the report $T_2$ of advertiser 2 is fixed. Let $N_k$ be an equilibrium that exists and recall that the fact that $N_k$ exists means we have a set of inequalities which are satisfied. Now consider a deviation of advertiser 1 to a lower report $T_1' < T_1$. If the inequalities establishing the existence of $N_k$ are satisfied then we are done as this trivially establishes that the new report has an equilibrium which is no better than the original report. If not, then our goal is to show that for some $k' < k$, we have that $N_{k'}$ exists. Our main technical argument is the following. Suppose that $N_{k'}$ does not exist. Then it must be that there is some inequality which is not satisfied. However, we will show that (i) this implies $k' \geq 1$ and (ii) that the same inequality

*Table 1.* Summary of main results (for SPA)

|  | non-uniform bidding | uniform bidding |
| --- | --- | --- |
| 2 advertisers | RAIC (Thm 1), OAIC (Thm 2) | RAIC (Thm 5), OAIC (Thm 6) |
| > 2 advertisers | RAIC (Thm 1), OAIC (Thm 2) | - |

is satisfied for $N_{k'-1}$. Therefore, by iterating all possible $k' \leq k$, we must end up at some $N_{k'}$ where the desired inequalities are satisfied. If no equilibrium $N_{k'}$ exists for any $1 \leq k' \leq k$, then we show $N_0$ must be an equilibrium.

**Conflict of Interest Disclosure** At the time of submission, authors C.L. and W.Z. were employed by Google Research. Google develops auto-bidding systems.

## 2. Related Works

The autobidding problem was initially formalized by Aggarwal et al. (2019). In their paper, they show that uniform bidding is a near-optimal bidding function in deterministic, truthful auctions. In the mechanism design literature, the *price of anarchy (PoA)* is a standard measure of efficiency which measures the ratio of the welfare (sum of advertisers' values) in an equilibrium compared with the optimal allocation. Aggarwal et al. (2019) further show that the PoA is 2 for SPA with ROS and budget constraints. There have since been a sequence of works that study the efficiency of other mechanisms. Mehta (2022) shows that randomization helps by designing a randomized, truthful mechanism that has a PoA strictly better than 2 for two bidders. Liaw et al. (2023) built upon and shows that using a randomized *non*-truthful mechanism further improves the PoA. The aforementioned two works assume bidders only specify an ROS constraint. Liaw et al. (2024) prove PoA results when autobidders have both an ROS and a budget constraint and study PoA results when the optimal allocation can or cannot be randomized. Liu & Shen (2023) also looks at bidding and efficiency in autobidding with both an ROS and a budget constraint. Deng et al. (2021); Balseiro et al. (2021a) study how one can incorporate machine-learned signals into the auction and specifically how to improve equilibrium efficiency using such signals. Recently, there has also been work on understanding interdependency in autobidding (Banchio et al., 2025).

Alimohammadi et al. (2023) introduces the concept of AIC and shows that advertisers often have an incentive to strategically misrepresent their high-level objectives to their auto-bidding agents to achieve better outcomes in FPA and SPA. The existence of multiple equilibria in games with autobidding agents presents a significant challenge for defining and analyzing incentive compatibility. Alimohammadi et al. (Alimohammadi et al., 2023) address this by comparing worst-

case truthful outcomes to best-case deviation outcomes. Our work further refines this by considering consistent equilibrium selection (worst-worst for RAIC, best-best for OAIC), allowing for a more direct ordinal preference modeling for advertisers with specific attitudes towards equilibrium ambiguity.

A related line of work looks at auction design where bidders may submit both values and constraints into the auction, particularly a ROS constraint (Balseiro et al., 2021b; 2022; 2024). However, these works assume that the constraints go directly into the mechanism. On the other hand, we assume that a bidder receives the constraints and it is only the bidder that directly interacts with the mechanism.

Feng et al. (2024) study PoA when advertisers are utility maximizers and strategically report their budgets. Li & Tang (2024) study computation complexity to find an approximate auto-bidding equilibrium, as well as the optimal revenue or welfare equilibria. Wang et al. (2024) studied the multiple rounds setting where each round is a single query, multiple slots auction. They proposed an AIC mechanism Coupled First-Price Auction (CFP) and a TIC (Time-Invariant Incentive Compatibility) mechanism Decoupled First-Price Auction (DFP). Last but not least, there is work studying the game that advertisers strategically submit constraints across multiple channels (Deng et al., 2023; Aggarwal et al., 2024b).

## 3. Preliminaries

Let $A$ be a set of advertisers and $Q$ be a set of $n$ queries. For each advertiser, there is an auto-bidder that bids on behalf of the advertiser. Each bidder $i \in A$ has value $v_{i,j}$ for query $j \in Q$. We let $b_{i,j}$ denote bidder $i$'s bid on query $j$. A single-slot auction is defined via an allocation function $\pi \colon \mathbb{R}_+^A to \{0,1\}^A$, where $\sum_{i \in A} \pi_i(b) = 1$ for all $b \in \mathbb{R}_+^A$, and a cost function $c \colon \mathbb{R}_+^A \to \mathbb{R}_+^A$ which denotes the price paid by advertiser $i$ if they are allocated the slot.

**Background on auto-bidding.** Suppose all bidders except $i$ have fixed their bids. Then the decision variables for bidder $i$ are $\{\pi_{i,j}\}_{j \in Q}$ where $\pi_{i,j}$ denotes whether bidder $i$ wins the query. We let $c_{i,j}$ denote the cost that advertiser $i$ must pay for query $j$ when they win.

The goal of the auto-bidder is to optimize the advertiser's total value subject to a tCPA constraint where advertiser

$i$'s total expected value is no less than their total expected spend. More formally, the auto-bidding agent for advertiser $i$ aims to solve the following optimization problem:

$$\text{maximize: } \sum_{j \in Q} \pi_{i,j} v_{i,j}$$

$$\text{subject to: } \sum_{j \in Q} \pi_{i,j} c_{i,j} \leq T_i \sum_{j \in Q} \pi_{i,j} v_{i,j} \quad \text{(tCPA)}$$

$$\forall j \in Q, \ \pi_{i,j} \in [0,1].$$

In the above optimization problem, $T_i$ represents bidder $i$'s tCPA constraint. With allocation $\pi$, $\forall i \in A$, we denote their total value as $\text{LW}_i(\pi) = T_i \sum_{j \in Q} \pi_{i,j} v_{i,j}$.

We adopt the assumption from previous work (Aggarwal et al., 2019; Deng et al., 2021; Liaw et al., 2023) that an auto-bidder $i$ bids at least $T_i \cdot v_{i,j}$ in a second price auction. This is because bidding $T_i \cdot v_{i,j}$ "obviously weakly dominates" any bid $b_{i,j} < T_i \cdot v_{i,j}$. [2]

**Assumption 1.** *Undominated bids assumption: In a second price auction, $\forall i \in A, j \in Q$ with reported target $T_i$, we assume $b_{i,j} \geq T_i \cdot v_{i,j}$.*

### 3.1. Auto-Bidder Equilibrium

We say that the bids $\{b_{i,j}\}$ are in an auto-bidder equilibrium if the following two statements holds for each bidder $i$:

1. Bidder $i$ satisfies their tCPA constraint: $\sum_{j \in Q} \pi_{i,j} c_{i,j} \leq T_i \sum_{j \in Q} \pi_{i,j} v_{i,j}$.

2. Bidder $i$ is stable. Let $\pi$ and $c$ be the resulting allocation and costs of $\{b_{i,j}\}$. Suppose bidder $i$ deviates to bids $\{b'_{i,j}\}_{j \in Q}$, while other bidders remain their bids in $\{b_{i,j}\}$. Let $\pi'$ and $c'$ denote the allocation and costs after bidder $i$'s deviation. Then either bidder $i$ does not gain more value, or bidder $i$ violates their constraint. Formally, at least one of the following two inequalities must hold:

   - $T_i \sum_{j \in Q} \pi'_{i,j} v_{i,j} \leq T_i \sum_{j \in Q} \pi_{i,j} v_{i,j}$
   - $\sum_{j \in Q} \pi'_{i,j} c_{i,j} > T_i \sum_{j \in Q} \pi'_{i,j} v_{i,j}$

Given an instance $S$ and a mechanism $\mathcal{M}$, let $\Pi^{\text{EQ}}$ denote the set of allocations with $\mathcal{M}$ at equilibrium. In the case of uniform bidding equilibrium, we constraint the bids $b_{i,j} = \alpha_i v_{i,j}$ for some $\alpha_i > 0$ and $b'_{i,j} = \alpha'_i v_{i,j}$ for some other $\alpha'_i > 0$. The remainder of the definition of an equilibrium remains unchanged.

---

[2]By "obviously weakly dominates," we mean that if the bidder bids $b_{i,j} < T_i \cdot v_{i,j}$ and wins, they can raise their bid to $T_i \cdot v_{i,j}$ and still win without changing their allocation or payment. Conversely, if they do not win, increasing their bid to at most $T_i \cdot v_{i,j}$ either leaves their allocation unchanged or strictly provides more value while keeping their tCPA constraint satisfied.

### 3.2. Auto-bidding advertiser incentive compatibility

We consider the following game. First, each advertiser reports a tCPA target to their own auto-bidder. Then, the auto-bidders bid and reach an auto-bidder equilibrium.

Assuming the advertisers are strategic, do they have incentive to report a tCPA target that is different from their true target? Consider advertiser $i \in A$ with target $T_i$. Suppose all other advertisers' reported targets are fixed. The auction mechanism $\mathcal{M}$ is also fixed. Let $\Pi(T_i)^{\text{EQ}}$ denote the set of allocations with $\mathcal{M}$ at auto-bidder equilibria with $i$'s reported target being $T_i$.

**Definition 1.** *Risk-averse Auto-bidding Incentive Compatibility (RAIC): An auction rule is RAIC if for any Advertiser $i \in A$ with a tCPA target $T_i$ and reported target $T'_i \leq T_i$:*

$$\min_{\pi \in \Pi(T_i)^{\text{EQ}}} \text{LW}_i(\pi) \geq \min_{\pi \in \Pi(T'_i)^{\text{EQ}}} \text{LW}_i(\pi).$$

**Definition 2.** *Optimistic Auto-bidding Incentive Compatibility (OAIC): An auction rule is OAIC if for any Advertiser $i \in A$ with a tCPA target $T_i$ and reported target $T'_i \leq T_i$:*

$$\max_{\pi \in \Pi(T_i)^{\text{EQ}}} \text{LW}_i(\pi) \geq \max_{\pi \in \Pi(T'_i)^{\text{EQ}}} \text{LW}_i(\pi).$$

Note that we restrict reported target to be at most the true target, because reporting a higher target could end up violating the advertiser's true target.

## 4. RAIC and OAIC in Second Price Auction

**Non-uniform bidding.** First, we study RAIC and OAIC for two advertisers in SPA with non-uniform bidding, i.e., there is no assumption that auto-bidders bid uniformly on each query. We show that SPA are both RAIC and OAIC for tCPA auto-bidders.

**Theorem 1.** *SPA is RAIC in the setting of non-uniform bidding tCPA auto-bidders.*

*Proof Sketch.* Without loss of generality, we focus on advertiser 1. For $i \neq 1$, we assume advertiser $i$'s reported target $T_i$ is fixed. Starting from the worst case equilibrium allocation $\pi$ for advertiser 1 when they report their true target $T_1$, we construct bids that form a new equilibrium in a scenario that advertiser 1 reporting $T'_1 < T_1$. The construction ensures advertiser 1 wins only a subset of queries they win in $\pi$, all the bidders meet their tCPA constraints, and no bidder can win additional queries without violating these constraints. $\square$

**Theorem 2.** *SPA is OAIC in the setting of non-uniform bidding tCPA auto-bidders.*

**Theorem 3.** *FPA is OAIC in the setting of non-uniform bidding tCPA auto-bidders.*

The proof for Theorem 1, 2, and 3 can be found in Appendix A.1, A.2, and A.3.

**Uniform bidding.** Next, and for the remainder of this section, we study RAIC and OAIC for two advertisers in SPA with uniform bidding. For each bidder $i \in A$ with a reported target $T_i$, we assume they have a uniform multiplier $\mu_i \geq 1$. Given this multiplier, we assume that $b_{i,j} = \mu_i T_i v_{i,j}$. Let us order the queries such that

$$\frac{v_{1,1}}{v_{2,1}} > \frac{v_{1,2}}{v_{2,2}} > \cdots > \frac{v_{1,n-1}}{v_{2,n-1}} > \frac{v_{1,n}}{v_{2,n}}. \tag{1}$$

We assume there is no tie among the ratios $\frac{v_{1,j}}{v_{2,j}}$. If there exist ties in the original input, we can always merge tied queries first as pre-processing. This step is for analysis only.

When $b_{1,j} = b_{2,j}$, we let bidder 1 win $j$. The tie breaking does not affect the conclusion in this paper. We include the proofs for tie breaking towards bidder 2 in the proofs for completeness. With bid multipliers $\mu_1$, $\mu_2$, there exists $k$, such that

$$\frac{T_1 v_{1,0}}{T_2 v_{2,0}} > \frac{T_1 v_{1,1}}{T_2 v_{2,1}} > \frac{T_1 v_{1,2}}{T_2 v_{2,2}} > \cdots > \frac{T_1 v_{1,k}}{T_2 v_{2,k}} \geq \frac{\mu_2}{\mu_1}$$
$$> \frac{T_1 v_{1,k+1}}{T_2 v_{2,k+1}} > \cdots > \frac{T_1 v_{1,n}}{T_2 v_{2,n}} > \frac{T_1 v_{1,n+1}}{T_2 v_{2,n+1}}, \tag{2}$$

where we added two "virtual" queries to handle two corner cases $\frac{\mu_2}{\mu_1} > \frac{T_1 v_{1,1}}{T_2 v_{2,1}}$ and $\frac{\mu_2}{\mu_1} < \frac{T_1 v_{1,n}}{T_2 v_{2,n}}$. Let $v_{1,0} = +\infty$, $v_{2,0} = 1$, $v_{1,n+1} = 1$, $v_{2,n+1} = +\infty$. Note that these are *not* real queries and the bidder does not receive value for winning query $0$ or $n + 1$.

When $\frac{\mu_2}{\mu_1}$ is in the range shown in Inequality (2), bidder 1 wins queries $\{1 \ldots k\}$ (empty set when $k = 0$), and bidder 2 wins queries $\{k + 1 \ldots n\}$ (empty set when $k = n$).

We define this allocation as $N_k$: an allocation $\pi = N_k$ with $k \in [0, n]$ if:

| | | |
|---|---|---|
| if $k = 0$, | $\forall j \in [1, n]$, | $\pi_{1,j} = 0, \pi_{2,j} = 1$ |
| if $k = n$, | $\forall j \in [1, n]$, | $\pi_{1,j} = 1, \pi_{2,j} = 0$ |
| if $k \in [1, n-1]$, | $\forall 1 \leq j \leq k$, | $\pi_{1,j} = 1, \pi_{2,j} = 0$ |
| if $k \in [1, n-1]$, | $\forall k+1 \leq j \leq n$, | $\pi_{1,j} = 0, \pi_{2,j} = 1$ |

In the uniform bidding setting, the allocation depends only on $\frac{\mu_2}{\mu_1}$, so any feasible allocation $\pi$ must be in $\{N_k : k \in [0, n]\}$.

We first characterize the condition that an auto-bidder equilibrium exists with allocation $N_k$ for reported targets $T_1, T_2$. Given $T_1, T_2$, there exists equilibrium EQ with allocation $N_k$ if and only if there exists $\mu_1, \mu_2$, such that:

1. $\frac{\mu_2}{\mu_1}$ falls between $\frac{T_1 v_{1,k+1}}{T_2 v_{2,k+1}}$ and $\frac{T_1 v_{1,k}}{T_2 v_{2,k}}$.

2. Each bidder satisfies their tCPA constraint.

3. Each bidder would violate their tCPA constraint if they raise bids to win one extra query.

4. $\mu_1 \geq 1, \mu_2 \geq 1$.

The following condition formalizes these constraints.

**Condition 1.** *With advertiser reported targets $T_1, T_2$, an auto-bidder equilibrium EQ with allocation $N_k$ exists if and only if there exist $\mu_1, \mu_2$ such that the all of the following conditions hold:*

$$\frac{T_1 v_{1,k+1}}{T_2 v_{2,k+1}} < \frac{\mu_2}{\mu_1} \leq \frac{T_1 v_{1,k}}{T_2 v_{2,k}}$$

$$\text{(Bidder 1 wins } \{1, \ldots, k\})$$

$$\text{(Bidder 2 wins } \{k+1, \ldots, n\})$$

$$\mu_1 \geq 1, \mu_2 \geq 1 \qquad \text{(Undominated bids)}$$

$$\mu_2 T_2 \sum_{j=1}^{k} v_{2,j} \leq T_1 \sum_{j=1}^{k} v_{1,j} \qquad \text{(bidder1 tCPA)}$$

$$\mu_1 T_1 \sum_{j=k+1}^{n} v_{1,j} \leq T_2 \sum_{j=k+1}^{n} v_{2,j} \qquad \text{(bidder2 tCPA)}$$

$$\mu_2 T_2 \sum_{j=1}^{k+1} v_{2,j} > T_1 \sum_{j=1}^{k+1} v_{1,j} \qquad \text{(bidder1 stable)}$$

$$\mu_1 T_1 \sum_{j=k}^{n} v_{1,j} > T_2 \sum_{j=k}^{n} v_{2,j} \qquad \text{(bidder2 stable)}$$

We first show a warm up proposition that only considers the constraints of one bidder. The proof of this proposition showcases the key idea behind the proofs for our main theorems. Recall for RAIC and OAIC, we require two conditions to be satisfied: the tCPA constraint and the "stableness" constraint. In the next proposition, we assume that bidder 2 does not respond and only consider the response of bidder 1 after advertiser 1 adjusts their target. The main proof will then need to also consider the response of bidder 2, which will be more complex since they may each respond to each other until they reach an equilibrium.

**Proposition 1.** *Consider two advertisers in SPA. Suppose with reported targets $(T_1, T_2)$, there exists bid multipliers $\mu_1, \mu_2$ to form allocation $N_k$ such that all the conditions*

*in Condition 1 are satisfied. Then with reported targets $(T_1' < T_1, T_2)$, there must exists $\mu_1'$, such that when the bid multipliers are $(\mu_1', \mu_2)$, the allocation is $N_{k'}$ where $k' \leq k$, and both Bidder 1's tCPA constraint and stableness constraint are satisfied.*

*Proof.* We first check the case when $\mu_1' = \mu_1$ and $k' = k$. Bidder 1's stableness constraint is still satisfied, because $T_2 \sum_{j=1}^{k+1} v_{2,j} > T_1 \sum_{j=1}^{k+1} v_{1,j}$ implies $T_2 \sum_{j=1}^{k+1} v_{2,j} > T_1' \sum_{j=1}^{k+1} v_{1,j}$ when $T_1' < T_1$.

However, the bidder's tCPA constraint might not be satisfied anymore because their target decreases. In this case, we know that:

$$\mu_2 T_2 \sum_{j=1}^{k} v_{2,j} > T_1' \sum_{j=1}^{k} v_{1,j} \tag{3}$$

Next, we find $\mu_1'$ such that $(\mu_1', \mu_2)$ forms an allocation $N_{k-1}$. The stableness condition for Bidder 1 with $N_{k-1}$ is exactly the same as Inequality (3), which means when $N_k$ breaks bidder 1's tCPA constraint, then $N_{k-1}$ must satisfy their stableness constraint. Indeed, being stable means they cannot win more queries without violating their tCPA constraint, so if they would violate their tCPA constraint at $N_k$, then they must be stable at $N_{k-1}$. With this argument, we iterate all $k' \in [0, k]$ decreasingly, until we find a $k'$ that satisfies both their tCPA and stable constraints. Note that if we get to $k' = 0$ or $\mu_1' = 1$, their tCPA constraint must be satisfied, and they must also satisfy their stableness constraint because they must fail the tCPA constraint with $k' + 1$ to reach $k'$. □

To simplify the notation, we define the set of queries $\{1, \ldots, k\}$ to be $L_k$ and $\{k, \ldots, n\}$ to be $R_k$. For advertiser $i$, we define their total value in $L_k$ and $R_k$ as:

$$V_i(L_k) = \sum_{j=1}^{k} v_{i,j} \quad \text{and} \quad V_i(R_k) = \sum_{j=k}^{n} v_{i,j}.$$

We state and show Lemma 1 below by the mediant property. This lemma will be used in the proofs for Theorem 4, Theorem 5, and Theorem 6.

**Lemma 1.** *With reported targets $T_1, T_2$, the following inequalities hold for all $\hat{k} \in Q$:*

$$\frac{v_{1,\hat{k}}}{v_{2,\hat{k}}} > \frac{V_1(R_{\hat{k}+1})}{V_2(R_{\hat{k}+1})} \quad \text{and} \quad \frac{V_1(L_{\hat{k}})}{V_2(L_{\hat{k}})} > \frac{v_{1,\hat{k}+1}}{v_{2,\hat{k}+1}}.$$

*Proof.* Since $\frac{v_{1,\hat{k}}}{v_{2,\hat{k}}} > \frac{v_{1,j}}{v_{2,j}}$ for all $j \in [\hat{k}+1, n]$, the mediant property implies that

$$\frac{v_{1,\hat{k}}}{v_{2,\hat{k}}} > \frac{\sum_{j=\hat{k}+1}^{n} v_{1,j}}{\sum_{j=\hat{k}+1}^{n} v_{2,j}} = \frac{V_1(R_{\hat{k}+1})}{V_2(R_{\hat{k}+1})}.$$

Similarly, since $\frac{v_{1,\hat{k}+1}}{v_{2,\hat{k}+1}} < \frac{v_{1,j}}{v_{2,j}}$ for all $j \in [1, \hat{k}]$, the mediant property gives

$$\frac{V_1(L_{\hat{k}})}{V_2(L_{\hat{k}})} = \frac{\sum_{j=1}^{\hat{k}} v_{1,j}}{\sum_{j=1}^{\hat{k}} v_{2,j}} > \frac{v_{1,\hat{k}+1}}{v_{2,\hat{k}+1}}. \quad □$$

Theorem 4 below simplifies Condition 1 by removing the bid multipliers. This simplification is crucial for the proof of our main conclusion in Theorem 5 and 6.

**Theorem 4.** *With advertiser reported targets $T_1$, $T_2$, Condition 1 is equivalent to the following statement. An auto-bidder equilibrium EQ with allocation $N_k$ exists if and only if all of the following conditions hold.*

$$\frac{T_1 V_1(L_{k+1})}{T_2 V_2(L_{k+1})} < \frac{v_{1,k} \cdot V_2(R_{k+1})}{v_{2,k} \cdot V_1(R_{k+1})} \tag{4}$$

$$\frac{T_1 V_1(R_k)}{T_2 V_2(R_k)} > \frac{v_{1,k+1} \cdot V_2(L_k)}{v_{2,k+1} \cdot V_1(L_k)} \tag{5}$$

$$\frac{T_2 V_2(R_{k+1})}{T_1 V_1(R_{k+1})} \geq 1 \tag{6}$$

$$\frac{T_1 V_1(L_k)}{T_2 V_2(L_k)} \geq 1 \tag{7}$$

The proof can be found in Appendix A.4. Theorem 4 simplifies Condition 1 by removing the bid multipliers. This simplification is crucial for the proof of our main conclusion in Theorem 5 and 6.

Next, we state Lemma 2 by the mediant inequality, which will be used to proof Lemma 3. The proof for Lemma 2 is in Appendix A.5

**Lemma 2.** *For $\forall \, 0 \leq \hat{k} < k$, the following two statements are both true:*

$$\frac{T_2 V_2(R_{\hat{k}})}{T_1 V_1(R_{\hat{k}})} < \frac{T_2 V_2(R_k)}{T_1 V_1(R_k)}, \quad \frac{T1 V_1(L_{\hat{k}})}{T_2 V_2(L_{\hat{k}})} < \frac{T1 V_1(L_k)}{T_2 V_2(L_k)}$$

To simplify Inequalities (6) and (7) in Theorem 4, with reported targets $T_1$, $T_2$, we define

$$\mathcal{K}_{min}(T_1, T_2) = \min\{k : \frac{T_2 V_2(R_{k+1})}{T_1 V_1(R_{k+1})} \geq 1\}$$

$$\mathcal{K}_{max}(T_1, T_2) = \max\{k : \frac{T_1 V_1(L_k)}{T_2 V_2(L_k)} \geq 1\}$$

**Lemma 3.** *With reported targets $T_1$, $T_2$, Inequality (6) and (7) in Theorem 4 both hold if and only if $\mathcal{K}_{min}(T_1, T_2) \leq k \leq \mathcal{K}_{max}(T_1, T_2)$.*

*Proof.* Based on the definition of $\mathcal{K}_{min}$ and $\mathcal{K}_{max}$, it is straightforward that if $k < \mathcal{K}_{min}(T_1, T_2)$, then Inequality (6) does not hold. If $k > \mathcal{K}_{max}(T_1, T_2)$, then Inequality (7) does not hold. So we have proved the "only if" part of the statement.

Next, by Lemma 2, $\forall k \geq k_{min}$ satisfies (6). $\forall k \leq k_{max}$ satisfies (7). Thus, if $\mathcal{K}_{min}(T_1, T_2) \leq k \leq \mathcal{K}_{max}(T_1, T_2)$, then Inequality (6) and (7) must both hold. $\square$

To prove SPA is RAIC, we show in Theorem 5 that if there exists an equilibrium with reported targets $(T_1, T_2)$ and allocation $N_k$, then there must exists an equilibrium with reported targets $(T_1' < T_1, T_2)$ and $N_{k'}$, such that $k' \leq k$. To better represent Theorem 4 with varying reported targets and $N_k$ value, we establish the following notations.

$$
\mathcal{C}_1(T_1, T_2, k) = \frac{v_{1,k} \cdot V_2(R_{k+1})}{v_{2,k} \cdot V_1(R_{k+1})} - \frac{T_1 V_1(L_{k+1})}{T_2 V_2(L_{k+1})},
$$
$$
\mathcal{C}_2(T_1, T_2, k) = \frac{T_1 V_1(R_k)}{T_2 V_2(R_k)} - \frac{v_{1,k+1} \cdot V_2(L_k)}{v_{2,k+1} \cdot V_1(L_k)}
$$

We get the following corollary by Theorem 4 and Lemma 3.

**Corollary 1.** *With advertiser reported targets $T_1$, $T_2$, Condition 1 is equivalent to: an auto-bidder equilibrium* EQ *with allocation $N_k$ exists if and only if the all of the following conditions hold:*

$$
\mathcal{C}_1(T_1, T_2, k) > 0, \quad \mathcal{C}_2(T_1, T_2, k) > 0,
$$
$$
\mathcal{K}_{min}(T_1, T_2) \leq k \leq \mathcal{K}_{max}(T_1, T_2).
$$

**Theorem 5.** *SPA is RAIC in the setting of two uniform bidding tCPA auto-bidders.*

*Proof.* Suppose advertiser 1's true target is $T_1$. Advertiser 2's reported target is $T_2$ and fixed. We prove the theorem by showing that with any reported target $T_1' < T_1$, for any auto-bidder equilibrium EQ and allocation $\pi = N_k$ with reported targets $(T_1, T_2)$, there exists an auto-bidder equilibrium EQ$'$ and allocation $\pi' = N_{k'}$ with reported targets $(T_1', T_2)$, such that $\mathrm{LW}_1(\pi') \leq \mathrm{LW}_1(\pi)$. Note that $\mathrm{LW}_1(\pi') \leq \mathrm{LW}_1(\pi)$ is equivalent to $k' \leq k$.

Because there exists EQ with allocation $\pi = N_k$ with reported targets $(T_1, T_2)$, we know that all the conditions in Corollary 1 are satisfied.

With reported targets $(T_1', T_2)$, for any $k' \leq k$, to show that there exists EQ$'$ and allocation $\pi' = N_{k'}$, we need to show that

$$
\mathcal{C}_1(T_1', T_2, k') > 0, \quad \mathcal{C}_2(T_1', T_2, k') > 0,
$$
$$
\mathcal{K}_{min}(T_1', T_2) \leq k' \leq \mathcal{K}_{max}(T_1', T_2).
$$

The following two lemmas describe some properties of $\mathcal{K}_{min}(T_1', T_2)$ and $\mathcal{K}_{max}(T_1', T_2)$.

**Lemma 4.** $\mathcal{C}_1(T_1', T_2, k') > 0$ *when* $k' = \mathcal{K}_{max}(T_1', T_2)$.

*Proof.* When $k' = \mathcal{K}_{max}(T_1', T_2)$, we know that $k' + 1$ does not satisfy Inequality (7) with reported targets $(T_1', T_2)$, so:

$$
\frac{T_1' V_1(L_{k'+1})}{T_2 V_2(L_{k'+1})} < 1
$$

The above inequality is equivalent to $\mathcal{C}_1(T_1', T_2, k') > 0$.

By Lemma 1,

$$
\frac{v_{1,k'} \cdot V_2(R_{k'+1})}{v_{2,k'} \cdot V_1(R_{k'+1})} > 1 > \frac{T_1' V_1(L_{k'+1})}{T_2 V_2(L_{k'+1})}
$$

$\square$

**Lemma 5.** $\mathcal{C}_2(T_1', T_2, k') > 0$ *when* $k' = \mathcal{K}_{min}(T_1', T_2)$.

*Proof.* When $k' = \mathcal{K}_{min}(T_1', T_2)$, we know that $k' - 1$ does not satisfy Inequality (6) with reported targets $(T_1', T_2)$, so $\frac{T_2 V_2(R_{k'})}{T_1' V_1(R_{k'})} < 1$.

By Lemma 1,

$$
\frac{v_{1,k'+1} \cdot V_2(L_{k'})}{v_{2,k'+1} \cdot V_1(L_{k'})} < 1 < \frac{T_1' V_1(R_{k'})}{T_2 V_2(R_{k'})}
$$

The above inequality is equivalent to $\mathcal{C}_2(T_1', T_2, k') > 0$.

$\square$

For $k' \leq k$ such that $N_{k'}$ is an equilibrium allocation with reported targets $(T_1', T_2)$, by Corollary 1, $k'$ must satisfy $\mathcal{K}_{min}(T_1', T_2) \leq k' \leq \mathcal{K}_{max}(T_1', T_2)$, thus $k' \in [\mathcal{K}_{min}(T_1', T_2), \min\{\mathcal{K}_{max}(T_1', T_2), k\}]$. In Lemma 6, we iterate all $k'$ in this range decreasingly, and show that there either exists an equilibrium allocation $N_{k'}$, or $\mathcal{C}_1(T_1', T_2, k') > 0$ and $\mathcal{C}_2(T_1', T_2, k') \leq 0$, which are used when we consider the case with $k' - 1$.

**Lemma 6.** *For $j \in [\mathcal{K}_{min}(T_1', T_2), \min\{\mathcal{K}_{max}(T_1', T_2), k\}]$, at least one of the following two statements must be true:*
*Statement 1, There exists $k' \in [j, \min\{\mathcal{K}_{max}(T_1', T_2), k\}]$, such that $\mathcal{C}_1(T_1', T_2, k') > 0$ and $\mathcal{C}_2(T_1', T_2, k') > 0$.*
*Statement 2, $\mathcal{C}_1(T_1', T_2, j) > 0$ and $\mathcal{C}_2(T_1', T_2, j) \leq 0$.*

*Proof.* We prove this lemma by induction.

**Base case:** $\quad j = \min\{\mathcal{K}_{max}(T_1', T_2), k\}.$

If $\mathcal{K}_{max}(T_1', T_2) \leq k$, $j = \mathcal{K}_{max}(T_1', T_2)$. By Lemma 4, $\mathcal{C}_1(T_1', T_2, j) > 0$.
If $\mathcal{K}_{max}(T_1', T_2) > k$, $j = k$, thus $\mathcal{C}_1(T_1, T_2, j) = \mathcal{C}_1(T_1, T_2, k) > 0$:

$$\mathcal{C}_1(T_1, T_2, j) = \frac{v_{1,j} \cdot V_2(R_{j+1})}{v_{2,j} \cdot V_1(R_{j+1})} - \frac{T_1 V_1(L_{j+1})}{T_2 V_2(L_{j+1})} > 0$$

$$\mathcal{C}_1(T_1', T_2, j) = \frac{v_{1,j} \cdot V_2(R_{j+1})}{v_{2,j} \cdot V_1(R_{j+1})} - \frac{T_1' V_1(L_{j+1})}{T_2 V_2(L_{j+1})}$$
$$> \frac{v_{1,j} \cdot V_2(R_{j+1})}{v_{2,j} \cdot V_1(R_{j+1})} - \frac{T_1 V_1(L_{j+1})}{T_2 V_2(L_{j+1})} > 0$$

The inequality above holds because $T_1' < T_1$. So we have proved that $\mathcal{C}_1(T_1', T_2, j) > 0$ for $j = \min\{\mathcal{K}_{max}(T_1', T_2), k\}$. If $\mathcal{C}_2(T_1', T_2, j) > 0$, then Statement 1 is true, otherwise Statement 2 is true, i.e., at least one of Statement 1 and 2 must be true.

**Inductive Step** Assume this lemma holds for $j \in (\mathcal{K}_{min}(T_1', T_2), \min\{\mathcal{K}_{max}(T_1', T_2), k\}]$, then we prove it also holds for $j - 1$. Assuming Statement 1 does not hold for $j - 1$, we show that $\mathcal{C}_1(T_1', T_2, j - 1) > 0$ and $\mathcal{C}_2(T_1', T_2, j - 1) \leq 0$, i.e., Statement 2 must hold for $j - 1$.

Assume Statement 1 does not hold for $j - 1$, which means there does not exist $k' \in [j - 1, \min\{\mathcal{K}_{max}(T_1', T_2), k\}]$, such that $\mathcal{C}_1(T_1', T_2, k') > 0$ and $\mathcal{C}_2(T_1', T_2, k') > 0$. Because $j \in [j - 1, \min\{\mathcal{K}_{max}(T_1', T_2), k\}]$, then it must be $\mathcal{C}_1(T_1', T_2, j) \leq 0$ or
$\mathcal{C}_2(T_1', T_2, j) \leq 0$. By the inductive assumption that the lemma holds for $j$, one of Statement 1 and 2 must hold, and we know that Statement 1 does not hold, so Statement 2 must hold for $j$, i.e., $\mathcal{C}_1(T_1', T_2, j) > 0$ and $\mathcal{C}_2(T_1', T_2, j) \leq 0$. Thus,

$$\mathcal{C}_2(T_1', T_2, j) = \frac{T_1' V_1(R_j)}{T_2 V_2(R_j)} - \frac{v_{1,j+1} \cdot V_2(L_j)}{v_{2,j+1} \cdot V_1(L_j)} \leq 0$$

$$\Rightarrow \quad \frac{T_1' V_1(L_j)}{T_2 V_2(L_j)} \leq \frac{v_{1,j+1} \cdot V_2(R_j)}{v_{2,j+1} \cdot V_1(R_j)}$$

$$\mathcal{C}_1(T_1', T_2, j - 1) = \frac{v_{1,j-1} \cdot V_2(R_j)}{v_{2,j-1} \cdot V_1(R_j)} - \frac{T_1' V_1(L_j)}{T_2 V_2(L_j)}$$
$$> \frac{v_{1,j+1} \cdot V_2(R_j)}{v_{2,j+1} \cdot V_1(R_j)} - \frac{T_1' V_1(L_j)}{T_2 V_2(L_j)} \geq 0$$

The above inequality holds because $\frac{v_{1,j+1}}{v_{2,j+1}} < \frac{v_{1,j-1}}{v_{2,j-1}}$. Remember we assume Statement 1 does not hold for $j - 1$, thus

at most one of $\mathcal{C}_1(T_1', T_2, j-1) > 0$ and $\mathcal{C}_2(T_1', T_2, j-1) > 0$ is true. And we have shown $\mathcal{C}_1(T_1', T_2, j - 1) > 0$, so it must be the case that $\mathcal{C}_1(T_1', T_2, j - 1) > 0$ and $\mathcal{C}_2(T_1', T_2, j - 1) \leq 0$, i.e., Statement 2 holds for $j - 1$.

$\square$

Finally, we are ready to finish the proof of Theorem 5. By Lemma 6, we know that one of Statement 1 and 2 must be true for $j = \mathcal{K}_{min}(T_1', T_2)$. By Lemma 5, Statement 2 is false for $j = \mathcal{K}_{min}(T_1', T_2)$, so Statement 1 must be true. Thus, we have proved that there exists an auto-bidder equilibrium with allocation $N_{k'}$ with reported targets $(T_1', T_2)$, such that $k' \leq k$.

$\square$

With similar techniques, we show SPA is OAIC for two uniform bidding auto-bidders; the proof can be found in Appendix A.6.

**Theorem 6.** *SPA is OAIC in the setting of two uniform bidding tCPA auto-bidders.*

# 5. Discussion and Future Work

In this paper, we refine the AIC definition in (Alimohammadi et al., 2023) by proposing two auto-bidding incentive compatibility concepts: RAIC and OAIC. We then analyze the incentive compatibility for advertisers with tCPA constraints in SPA. For both RAIC and OAIC, we can derive ordinal preferences regarding the reporting of different constraints. Based on these preferences, we define auction mechanism incentive compatibility in terms of advertisers consistently preferring their true constraints. For the uniform bidding with two advertisers setting, we establish tools by establishing an equivalent condition for the existence of an equilibrium allocation where one bidder wins a specific number of queries. The generalization of our analysis to more than two bidders under uniform bidding is left for future research, as it presents complexities beyond a straightforward extension of the two-advertiser case. However, we anticipate that the techniques presented in this paper will serve as valuable tools for future work in this area.

While a significant body of recent literature has focused on designing novel auction mechanisms to address the incentive issues arising from auto-bidders, our findings offer a more optimistic view of existing systems. Previous work (Alimohammadi et al., 2023) suggested that standard auctions like SPA might suffer from severe incentive vulnerabilities when advertisers delegate bidding to agents with tCPA constraints. However, by evaluating incentives through the refined lenses of RAIC and OAIC, our results indicate that SPA actually exhibits reasonable incentive alignment. A key practical implication of this work is that ad platforms may not necessarily need to radically revamp their existing

auction architectures to account for strategic auto-bidding behaviors.

Another closely related direction is to study advertisers with budget constraints. In settings where both tCPA and budget constraints are present, our results apply when the tCPA constraint is the binding one. For scenarios with budget constraints but no tCPA constraints, the undominated bids assumption does not hold for SPA. This is because a bidder could bid less than their value on queries to avoid violating their budget constraints. Without any bid assumptions, the optimal equilibrium for an advertiser is trivially to win all queries by bidding infinitely, while all others bid zero. Conversely, the worst-case equilibrium is winning nothing. Notably, these outcomes are independent of the advertisers' stated budgets, making the budget-only scenario less compelling in SPA. Furthermore, regarding equilibrium selection, because SPA satisfies both the extreme worst-case (RAIC) and best-case (OAIC) boundaries, it naturally satisfies any convex combination of the two, meaning these results hold for advertisers with "mixed" or moderate attitudes toward equilibrium ambiguity.

## Impact Statement

This paper presents work whose goal is to advance the field of algorithmic game theory and mechanism design for online advertising. Because our contributions are fundamentally theoretical in nature, we do not foresee any immediate societal consequences.

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

## A. Missing proofs in Section 4

### A.1. Proof of Theorem 1

**Theorem 1 (Restated)** *SPA is RAIC in the setting of non-uniform bidding tCPA auto-bidders.*

*Proof.* Without loss of generality, we focus on advertiser 1. Let $T_i$ denote the target of each advertiser. For $i \neq 1$, we assume advertiser $i$'s target is fixed.

Let $\pi = \text{argmin}_{\pi \in \Pi(T_1)^{\text{EQ}}} \text{LW}_i(\pi)$, i.e. $\pi$ is the worst case equilibrium for advertiser 1 when they report their true target $T_1$. If there are multiple equilibria that have the same worst case welfare for advertiser 1, we take an arbitrary one as $\pi$. For any reported target $T_1' < T_1$, we show that there exists $\pi' \in \Pi(T_1')^{\text{EQ}}$, such that $\text{LW}_1(\pi') \leq \text{LW}_1(\pi)$, which indicates $\min_{\pi \in \Pi(T_1^{\text{EQ}}} \text{LW}_1(\pi) \geq \min_{\pi \in \Pi(T_1')^{\text{EQ}}} \text{LW}_1(\pi)$. This implies that the auction mechanism is RAIC.

Since $\pi$ is an auto-bidder equilibrium, we know that all the bidders satisfy their tCPA constraints and that no bidder can increase its welfare from deviating from its current bids. In addition, we assume all bidders satisfy the undominated bids assumption so $b_{i,j} \geq T_i v_{i,j}$ for all $i \in A$ and $j \in Q$. We now construct bids that form a new equilibrium $\pi'$ when advertiser 1 reports $T_1' < T_1$ and $T_i' = T_i$ for $i \neq 1$.

For $j \in Q$, let $w(j)$ denote the winner that wins $j$ in $\pi$. For queries where bidder 1 loses the auction in $\pi$, we will keep bidder 1's bids and increase the original winner's bids, so that the allocation on these queries are the same in $\pi$ and $\pi'$. Formally, for $j \in Q$ such that $w(j) \neq 1$, let $b'_{w(j),j} = +\infty$, and for $i \neq w(j)$, let $b'_{i,j} = b_{i,j}$. It is straightforward that $\pi'_{i,j} = \pi_{i,j}$ for all $i \in A$ for such queries.

Now, fix query $j$ where bidder 1 did win the auction in $\pi$. Let $w^*(j) = \text{argmax}_{i \neq 1} T_i' v_{i,j}$. We consider two cases.

**Case 1:** $T_1' v_{1,j} \geq T_{w^*(j)}' v_{w^*(j),j}$. In this case, let $b'_{1,j} = +\infty$ and for $i \neq 1$, set $b'_{i,j} = T_i' v_{i,j}$. As a result, bidder 1 wins $j$ in $\pi'$.

**Case 2:** $T_1' v_{1,j} < T_{w^*(j)}' v_{w^*(j),j}$. In this case, let $b'_{w^*(j),j} = +\infty$. For all $i \neq w^*(j)$, set $b'_{i,j} = T_i' v_{i,j}$. As a result, bidder $w^*(j)$ wins $j$ in $\pi'$.

Next, we show that $b'_{i,j}$ form an auto-bidder equilibrium where bidder 1 reports $T_1' < T_1$ and bidder $i \neq 1$ reports $T_i' = T_i$. First, we show that all the bids in $b'_{i,j}$ satisfy the undominated bids assumption. For queries $j \in Q$ where bidder 1 was not winning in $\pi$, we have $b'_{i,j} \geq b_{i,j}$ and we know that $b_{i,j} \geq T_i v_{i,j} \geq T_i' v_{i,j}$. For queries $j \in Q$ where bidder 1 was winning in $\pi$, the undominated bids assumption is trivially satisfied by construction. Second,

observe that no bidder can win more queries than what they win in $\pi'$, because for the queries they lose, the other bidder bids $+\infty$.

We now check that the tCPA constraints are satisfied. Bidder 1 satisfies their tCPA constraint, because for all the queries they win in $\pi'$, the cost, or second-highest bid, is at most $T_1' v_{1,j}$. Similarly, all other bidders satisfies their tCPA constraint, because for all the queries they win in both $\pi$ and $\pi'$, we know the constraint is satisfied because $\pi$ forms an equilibrium with reported targets $(T_1, T_2, \ldots, T_n)$. Moreover, the cost for these bidders are the same in $\pi$ and $\pi'$. For the queries bidders other than bidder 1 win in $\pi'$ but not $\pi$ (Case 2 above), we know that the cost is at most $T_{w^*(j)}' v_{w^*(j),j}$, so the constraint remains satisfied.

Finally, we show that $\mathrm{LW}_1(\pi) \leq \mathrm{LW}_1(\pi')$. This is simply because if bidder 1 loses a query in $\pi$ then it continues to lose in $\pi'$ by construction. □

## A.2. Proof of Theorem 2

**Theorem 2 (Restated)** *SPA is OAIC in the setting of non-uniform bidding tCPA auto-bidders.*

*Proof.* Without loss of generality, suppose advertiser 1's true target is $T_1$, any other Advertiser $i$'s reported target is $T_i$ and fixed.

For any reported target $T_1' < T_1$, and $\forall i \neq 1$, $T_i' = T_i$. let $\pi' = \mathrm{argmax}_{\pi \in \Pi(T_1')^{\mathrm{EQ}}} \mathrm{LW}_i(\pi)$, i.e. $\pi'$ is the best case equilibrium for advertiser 1 when they report target $T_1'$. If there are multiple equilibria that have the same best case welfare for advertiser 1, we take an arbitrary one as $\pi'$. We show that when advertiser 1 reports their true target $T_1$, there exists $\pi \in \Pi(T_1)^{\mathrm{EQ}}$, such that $\mathrm{LW}_1(\pi') \leq \mathrm{LW}_1(\pi)$, which indicates $\max_{\pi \in \Pi(T_1)^{\mathrm{EQ}}} \mathrm{LW}_1(\pi) \geq \max_{\pi \in \Pi(T_1')^{\mathrm{EQ}}} \mathrm{LW}_1(\pi)$, thus the auction mechanism is OAIC.

When advertiser 1 reports $T_1'$, in the auto-bidder equilibrium with allocation $\pi'$, suppose $\forall i \in A, j \in Q$ bidder i's bid is $b_{i,j}'$. By the definition of an auto-bidder equilibrium, we know that all the bidders satisfy their tCPA constraints, and no bidder could increase their welfare by deviating from their current bids. Also all the bidders satisfy the undominated bids assumption: $\forall i \in A, j \in Q, b_{i,j}' \geq T_i' v_{i,j}$.

Now we construct bids $b$ that form an equilibrium $\pi$ when advertiser 1 reports $T_1 > T_1'$ and $\forall i \neq 1, T_i = T_i'$.

$\forall j \in Q$, let $w(j)$ denote the winner that wins $j$ in $\pi'$. For queries that bidder 1 wins the auction in $\pi'$, we increase bidder 1's bids and keep all other bidders' bids in $\pi'$, which keeps the same allocation as in $\pi'$. Formally, $\forall j \in Q$ such that $w(j) = 1$, let $b_{1,j} = +\infty$, and for $i \neq 1$, let $b_{i,j} = b_{i,j}'$. It is straight-forward that $\forall i \in A, \pi_{i,j} = \pi_{i,j}'$ for these queries.

For query $j$ that bidder 1 loses the auction in $\pi'$, let $w^*(j) = \mathrm{argmax}_{i \neq 1} T_i v_{i,j}$. we group them to two cases:

**Case 1:** $T_1 v_{1,j} \geq T_{w^*(j)} v_{w^*(j),j}$. In this case, let $b_{1,j} = +\infty$ and $\forall i \neq 1, b_{i,j} = T_i v_{i,j}$. As a result, bidder 1 wins $j$ in $\pi$.

**Case 2:** $T_1 v_{1,j} < T_{w^*(j)} v_{w^*(j),j}$. In this case, let $b_{w^*(j),j} = +\infty$, and $\forall i \neq w^*(j), b_{i,j} = T_i v_{i,j}$. As a result, bidder $w^*(j)$ wins $j$ in $\pi$.

Next, we show that $b_{i,j}$ form an auto-bidder equilibrium with reported targets $(T_1 > T_1', T_2 = T_2', \ldots, T_n = T_n')$. First, all the bids in $b_{i,j}$ satisfy the undominated bids assumption. For bidders $i \neq 1$, $b_{i,j} \geq b_{i,j}'$. The bids satisfy the undominated bids assumption because bids $b_{i,j}$ satisfy the assumption and their targets are fixed. For advertiser 1, $b_{1,j} = +\infty$ or $b_{1,j} = T_1 v_{1,j}$, which also satisfy the undominated bids assumption in both cases. Second, no bidder could win more queries than what they win in $\pi$, because for the queries they lose, the winner bids $+\infty$.

$\forall i \neq 1$, bidder $i$ satisfies their tCPA constraint, because the cost for query $j$ is less than $T_i v_{i,j}$ (Case 2 above). Bidder 1 also satisfies their tCPA constraint: For all the queries that bidder 1 wins in both $\pi'$ and $\pi$, the tCPA constraint is satisfied, because $\pi'$ forms an equilibrium with reported targets $(T_1' < T_1, T_2' = T_2, \ldots, T_n' = T_n)$, the cost for bidder 1 is the same in $\pi'$ and $\pi$, and $T_1 > T_1'$. For the queries that bidder 1 loses in $\pi'$ but wins in $\pi$ (Case 1), we know that the cost $T_{w^*(j)} v_{w^*(j),j} < T_1 v_{1,j}$, so the constraint is also satisfied.

Finally, we show that $\mathrm{LW}_1(\pi') \leq \mathrm{LW}_1(\pi)$: For any query bidder 1 wins in $\pi'$, they still wins in $\pi$, thus $\mathrm{LW}_1(\pi') \leq \mathrm{LW}_1(\pi)$. □

## A.3. Proof of Theorem 3

**Theorem 3 (Restated)** *FPA is OAIC in the setting of non-uniform bidding tCPA auto-bidders.*

First, we require the following observation.

**Claim 1.** *Consider the setting of non-uniform bidding tCPA bidders under FPA. In any equilibrium, on any query, the winning bidder pays at least the value of any losing bidder.*

*Proof.* Fix a query and let $v$ denote the value of a losing bidder. If the winner pays (and bids) less than $v$ then the losing bidder can bid $v$ to win the query without violating its slack. Thus the winner must be paying and bidding at least $v$ in any equilibrium. □

*Proof.* Without loss of generality, we consider bidder (and advertiser) 1. By Claim 1, we know that if bidder 1 wishes

to win query $j$ then they must pay at least $\max_{i \neq 2} v_{i,j}$. Let $p_j = \max_{i \neq 2} v_{i,j}$ be the price bidder 1 must pay if they wish to win query $j$. First, suppose that the advertiser submits their Thus, an upper bound on bidder 1's value is the solution to the optimization problem

$$\text{maximize} \quad \sum_j T_1 v_{1,j} x_j$$

$$\text{subject to} \quad \sum_j (T_1 v_{1,j} - p_j) x_j \geq 0$$

$$x_j \in \{0, 1\}.$$

Let $S$ denote an optimal solution to the above optimization problem, i.e. $S = \{j : x_j = 1\}$. We claim that $\sum_{j \in S} v_{1,j}$ is the highest value that advertiser 1 can obtain over all possible equilibria.

To that end, it suffices to show that the following is an autobidder equilibrium. For a query $j \in S$, bidder 1 places a bid of $p_j$. Every other bidder either places a bid of $p_j$ (if it comes after bidder 1 in the tie-breaking order) or $p_j - 1$ (it it comes before bidder 1 in the tie-breaking order). For a query $j \notin S$, all bidders bid their value. Bidder 1 bids $p_j$ if it comes after the highest value bidder in the tie-breaking order or $p_j - 1$ otherwise.

First, observe that with these bids, bidder 1 obtains exactly $S$. Bidder 1 cannot deviate because if it did, this would find a solution even better than $S$ which is impossible. For every other bidder, their constraint is exactly tight and they are paying their value for every query. On every query, they must pay strictly more than their value. Thus, it is infeasible for them to win any query that they are currently not winning.

Now assume advertiser 1 lowers its target from $T_1$ to $T_1'$. Now their optimal value is obtained by solving the optimization problem

$$\text{maximize} \quad \sum_j T_1 v_{1,j} x_j$$

$$\text{subject to} \quad \sum_j (T_1' v_{1,j} - p_j) x_j \geq 0$$

$$x_j \in \{0, 1\}.$$

Let $S'$ denote a solution to this optimization problem.

Following the above argument above, the highest value that advertiser 1 can obtain over all possible autobidder equilibria is $\sum_{j \in S'} v_{1,j}$.

Now observe that $S'$ is also a feasible solution to the original optimization problem when the advertiser submitted its true target $T_1$. This is because $T_1' \leq T_1$ so we know that $\sum_{j \in S'} (T_1 v_{1,j} - p_j) \geq \sum_{j \in S'} (T_1' v_{1,j} - p_j) \geq 0$. Thus, we conclude that $\sum_{j \in S} T_1 v_{1,j} \geq \sum_{j \in S'} T_1 v_{1,j}$, which in turn concludes that FPA is OAIC. □

## A.4. Proof of Theorem 4

*Proof of Theorem 4.* Rewriting Condition 1 using $L_k$ and $R_k$, we have the following conditions.

$$\frac{T_1 v_{1,k+1}}{T_2 v_{2,k+1}} < \frac{\mu_2}{\mu_1} \leq \frac{T_1 v_{1,k}}{T_2 v_{2,k}} \tag{8}$$

$$\mu_2 T_2 V_2(L_k) \leq T_1 V_1(L_k) \tag{9}$$

$$\mu_1 T_1 V_1(R_{k+1}) \leq T_2 V_2(R_{k+1}) \tag{10}$$

$$\mu_2 T_2 V_2(L_{k+1}) > T_1 V_1(L_{k+1}) \tag{11}$$

$$\mu_1 T_1 V_1(R_k) > T_2 V_2(R_k) \tag{12}$$

$$\mu_1 \geq 1, \mu_2 \geq 1 \tag{13}$$

We first show the equivalent conditions for different subsets of Condition 1 and then combine them to finish the proof.

**Lemma 7.** *There exist $\mu_1, \mu_2$ to satisfy Inequality* (9), (10), (11), (12), (13) *if and only if the inequalities below all hold:*

$$\left.\begin{array}{l} \text{If } \dfrac{T_2 V_2(R_k)}{T_1 V_1(R_k)} < 1, \quad \dfrac{\mu_2}{\mu_1} \leq \dfrac{T_1 V_1(L_k)}{T_2 V_2(L_k)} \\[2ex] \text{If } \dfrac{T_2 V_2(R_k)}{T_1 V_1(R_k)} \geq 1, \quad \dfrac{\mu_2}{\mu_1} < \dfrac{T_1 V_1(L_k) T_1 V_1(R_k)}{T_2 V_2(L_k) T_2 V_2(R_k)} \\[2ex] \text{If } \dfrac{T_1 V_1(L_{k+1})}{T_2 V_2(L_{k+1})} < 1, \quad \dfrac{\mu_2}{\mu_1} \geq \dfrac{T_1 V_1(R_{k+1})}{T_2 V_2(R_{k+1})} \\[2ex] \text{If } \dfrac{T_1 V_1(L_{k+1})}{T_2 V_2(L_{k+1})} \geq 1, \quad \dfrac{\mu_2}{\mu_1} > \dfrac{T_1 V_1(L_{k+1}) T_1 V_1(R_{k+1})}{T_2 V_2(L_{k+1}) T_2 V_2(R_{k+1})} \end{array}\right\} \tag{14}$$

$$\frac{T_2 V_2(R_{k+1})}{T_1 V_1(R_{k+1})} \geq 1 \tag{6}$$

$$\frac{T_1 V_1(L_k)}{T_2 V_2(L_k)} \geq 1 \tag{7}$$

*Proof.* The feasible ranges of $\mu_1, \mu_2$ from Inequalities (9), (10), (11), (12) are equivalent to:

$$\frac{T_2 V_2(R_k)}{T_1 V_1(R_k)} < \mu_1 \leq \frac{T_2 V_2(R_{k+1})}{T_1 V_1(R_{k+1})}$$

$$\frac{T_1 V_1(L_{k+1})}{T_2 V_2(L_{k+1})} < \mu_2 \leq \frac{T_1 V_1(L_k)}{T_2 V_2(L_k)} \tag{15}$$

We combine the above conditions with Inequality (13). Inequality (6) and (7) are necessary for the ranges in Inequality (13) and (15) to overlap. Iterating different cases, there exist $\mu_1, \mu_2$ that satisfy both Inequality (15) and (13) if and only if Inequality (7), (6), (7) all hold. □

**Lemma 8.** *There exists feasible $\mu_1, \mu_2$ that satisfies Inequalities* (8) *and* (7), *if and only if Inequality* (4) *and* (5) *both hold.*

*Proof.* There exists feasible $\mu_1$, $\mu_2$ that satisfies Inequalities (8) and (7), if and only if the range of $\frac{\mu_2}{\mu_1}$ in (8) and (7) have overlap.

We split (8) into two cases:

**Case 1:** $\frac{T_1 v_{1,k+1}}{T_2 v_{2,k+1}} < \frac{\mu_2}{\mu_1} < \frac{T_1 v_{1,k}}{T_2 v_{2,k}}$. In this case, $\frac{T_1 v_{1,k+1}}{T_2 v_{2,k+1}} < \frac{\mu_2}{\mu_1} < \frac{T_1 v_{1,k}}{T_2 v_{2,k}}$ and (7) overlap is equivalent to:

$$\frac{T_1 v_{1,k}}{T_2 v_{2,k}} > \frac{T_1 V_1(R_{k+1})}{T_2 V_2(R_{k+1})} \cdot \max\{\frac{T_1 V_1(L_{k+1})}{T_2 V_2(L_{k+1})}, 1\}$$

$$\frac{T_1 V_1(L_k)}{T_2 V_2(L_k)} \cdot \min\{\frac{T_1 V_1(R_k)}{T_2 V_2(R_k)}, 1\} > \frac{T_1 v_{1,k+1}}{T_2 v_{2,k+1}}$$

Cancel $\frac{T_1}{T_2}$ on both sides:

$$\frac{v_{1,k}}{v_{2,k}} > \frac{V_1(R_{k+1})}{V_2(R_{k+1})} \cdot \max\{\frac{T_1 V_1(L_{k+1})}{T_2 V_2(L_{k+1})}, 1\}$$

$$\frac{V_1(L_k)}{V_2(L_k)} \cdot \min\{\frac{T_1 V_1(R_k)}{T_2 V_2(R_k)}, 1\} > \frac{v_{1,k+1}}{v_{2,k+1}} \quad (16)$$

**Case 2.1: Tie breaking towards bidder 1 and** $\frac{T_1 v_{1,k+1}}{T_2 v_{2,k+1}} < \frac{\mu_2}{\mu_1} = \frac{T_1 v_{1,k}}{T_2 v_{2,k}}$**.** We assume bidder 1 wins when there is a tie in bids. The only scenario in this case not covered by the strict conditions in Inequality (16) is the third line in Inequality (7) when $\frac{T_1 V_1(L_{k+1})}{T_2 V_2(L_{k+1})} < 1$ and $\frac{\mu_2}{\mu_1} = \frac{T_1 V_1(R_{k+1})}{T_2 V_2(R_{k+1})}$. Thus, for $\frac{T_1 v_{1,k+1}}{T_2 v_{2,k+1}} < \frac{\mu_2}{\mu_1} = \frac{T_1 v_{1,k}}{T_2 v_{2,k}}$ and (7) to have overlap is equivalent to either Inequality (16) holds, or both the following two inequalities hold:

$$\frac{T_1 V_1(L_{k+1})}{T_2 V_2(L_{k+1})} < 1, \quad \frac{\mu_2}{\mu_1} = \frac{T_1 v_{1,k}}{T_2 v_{2,k}} = \frac{T_1 V_1(R_{k+1})}{T_2 V_2(R_{k+1})} \quad (17)$$

By Lemma 1, Inequality (17) in Case 2.1 does not hold, so this scenario is ruled out.

**Case 2.2: Tie breaking towards bidder 2 and** $\frac{T_1 v_{1,k+1}}{T_2 v_{2,k+1}} = \frac{\mu_2}{\mu_1} < \frac{T_1 v_{1,k}}{T_2 v_{2,k}}$**.** We assume bidder 2 wins when there is a tie in bids. The only scenario in this case not covered by the strict conditions in Inequality (16) is the first line in Inequality (7) when $\frac{T_2 V_2(R_k)}{T_1 V_1(R_k)} < 1$ and $\frac{\mu_2}{\mu_1} = \frac{T_1 V_1(L_k)}{T_2 V_2(L_k)}$. Thus, for $\frac{T_1 v_{1,k+1}}{T_2 v_{2,k+1}} = \frac{\mu_2}{\mu_1} < \frac{T_1 v_{1,k}}{T_2 v_{2,k}}$ and (7) to have overlap is equivalent to either Inequality (16) holds, or both the following two inequalities hold:

$$\frac{T_2 V_2(R_k)}{T_1 V_1(R_k)} < 1, \quad \frac{\mu_2}{\mu_1} = \frac{T_1 v_{1,k+1}}{T_2 v_{2,k+1}} = \frac{T_1 V_1(L_k)}{T_2 V_2(L_k)} \quad (18)$$

By Lemma 1, Inequality (18) in Case 2.2 does not hold, so this scenario is ruled out.

Unifying Case 1 and Case 2, both (8) and (7) hold if and only if Inequality (16) holds.

Using Lemma 1, Inequality (16) could be simplified to (4) and (5).

$\square$

Now we are ready to finish the proof of Theorem 4. Combining Lemma 7 and Lemma 8, there exists $\mu_1$, $\mu_2$ that all the inequalities in Condition 1 are satisfied if and only if Inequality (4), (5), (6), (7) all hold.

$\square$

**A.5. Proof of Lemma 2**

**Lemma 2 (Restated)** *For $\forall\, 0 \le \hat{k} < k$, the following two statements are both true:*

$$\frac{T_2 V_2(R_{\hat{k}})}{T_1 V_1(R_{\hat{k}})} < \frac{T_2 V_2(R_k)}{T_1 V_1(R_k)}, \quad \frac{T1 V_1(L_{\hat{k}})}{T_2 V_2(L_{\hat{k}})} < \frac{T1 V_1(L_k)}{T_2 V_2(L_k)}$$

*Proof.* Remember the queries are ordered by Inequality (1). $\forall j \in [k, n]$, by the mediant inequality, we know that

$$\frac{T_2 v_{2,j}}{T_1 v_{1,j}} > \frac{T_2 v_{2,k-1}}{T_1 v_{1,k-1}} \ge \frac{T_2 \sum_{j=\hat{k}}^{k-1} v_{2,j}}{T_1 \sum_{j=\hat{k}}^{k-1} v_{1,j}}$$

Again by the mediant inequality,

$$\frac{T_2 V_2(R_{\hat{k}})}{T_1 V_1(R_{\hat{k}})} = \frac{T_2 \sum_{j=\hat{k}}^{n} v_{2,j}}{T_1 \sum_{j=\hat{k}}^{n} v_{1,j}} = \frac{T_2(\sum_{j=\hat{k}}^{k-1} v_{2,j} + \sum_{j=k}^{n} v_{2,j})}{T_1(\sum_{j=\hat{k}}^{k-1} v_{1,j} + \sum_{j=k}^{n} v_{1,j})}$$

$$< \frac{T_2 \sum_{j=k}^{n} v_{2,j}}{T_1 \sum_{j=k}^{n} v_{1,j}} = \frac{T_2 V_2(R_k)}{T_1 V_1(R_k)}$$

Similarly, $\forall j \in [\hat{k}+1, k]$, by the mediant inequality, we know that

$$\frac{T_1 v_{1,j}}{T_2 v_{2,j}} < \frac{T_1 v_{1,\hat{k}-1}}{T_2 v_{2,\hat{k}-1}} \le \frac{T_1 \sum_{j=1}^{\hat{k}} v_{1,j}}{T_2 \sum_{j=1}^{\hat{k}} v_{2,j}}$$

By the mediant inequality, we bound $\frac{T_1 V_1(L_k)}{T_2 V_2(R_k)}$ as follows:

$$\frac{T_1 \sum_{j=1}^{k} v_{1,j}}{T_2 \sum_{j=1}^{k} v_{2,j}} = \frac{T_1(\sum_{j=1}^{\hat{k}} v_{1,j} + \sum_{j=\hat{k}+1}^{k} v_{1,j})}{T_2(\sum_{j=1}^{\hat{k}} v_{2,j} + \sum_{j=\hat{k}+1}^{k} v_{2,j})}$$

$$< \frac{T_1 \sum_{j=1}^{\hat{k}} v_{1,j}}{T_2 \sum_{j=1}^{\hat{k}} v_{2,j}} = \frac{T1 V_1(L_{\hat{k}})}{T_2 V_2(L_{\hat{k}})}$$

$\square$

### A.6. Proof of Theorem 6

**Theorem 6 (Restated)** *SPA is OAIC in the setting of two uniform bidding tCPA auto-bidders.*

*Proof.* Suppose advertiser 1's true target is $T_1$. Advertiser 2's reported target is $T_2$ and fixed. We prove the theorem by showing that with any reported target $T_1' < T_1$, for any auto-bidder equilibrium EQ' and allocation $\pi' = N_{k'}$ with reported targets $(T_1', T_2)$, there exists an auto-bidder equilibrium EQ and allocation $\pi = N_k$ with reported targets $(T_1, T_2)$, such that $\text{LW}_1(\pi') \leq \text{LW}_1(\pi)$. Note that $\text{LW}_1(\pi') \leq \text{LW}_1(\pi)$ is equivalent to $k' \leq k$. The rest of the proof is similar to Theorem 5.

Because there exists EQ' with allocation $\pi' = N_{k'}$ with reported targets $(T_1', T_2)$, by Corollary 1, we know that

$$\mathcal{C}_1(T_1', T_2, k') > 0$$
$$\mathcal{C}_2(T_1', T_2, k') > 0$$
$$\mathcal{K}_{min}(T_1', T_2) \leq k' \leq \mathcal{K}_{max}(T_1', T_2)$$

With reported targets $(T_1, T_2)$, for any $k \geq k'$, to show that there exists EQ and allocation $\pi = N_k$, we need to show:

$$\mathcal{C}_1(T_1, T_2, k) > 0$$
$$\mathcal{C}_2(T_1, T_2, k) > 0$$
$$\mathcal{K}_{min}(T_1, T_2) \leq k \leq \mathcal{K}_{max}(T_1, T_2)$$

Similar to Lemma 6, for $k \geq k'$ such that $N_k$ is an equilibrium allocation with reported targets $(T_1, T_2)$, by Corollary 1, $k$ must satisfy $\mathcal{K}_{min}(T_1, T_2) \leq k \leq \mathcal{K}_{max}(T_1, T_2)$, thus $k \in [\max\{\mathcal{K}_{min}(T_1, T_2), k'\}, \mathcal{K}_{max}(T_1, T_2)]$. In Lemma 9, we iterate all $k$ in this range increasingly, and show that there either exists an equilibrium allocation $N_k$, or $\mathcal{C}_1(T_1, T_2, j) \leq 0$ and $\mathcal{C}_2(T_1, T_2, j) > 0$, which are useful when we consider the case with $k + 1$.

**Lemma 9.** *For each $j \in [\max\{\mathcal{K}_{min}(T_1, T_2), k'\}, \mathcal{K}_{max}(T_1, T_2)]$, at least one*

*of the following two statements must be true:*
*Statement 1, There exists $k \in [\max\{\mathcal{K}_{min}(T_1, T_2), k'\}, j]$, such that $\mathcal{C}_1(T_1, T_2, k) > 0$ and $\mathcal{C}_2(T_1, T_2, k) > 0$.*
*Statement 2, $\mathcal{C}_1(T_1, T_2, j) \leq 0$ and $\mathcal{C}_2(T_1, T_2, j) > 0$.*

*Proof.* We prove this lemma by induction.

**Base case** : $j = \max\{\mathcal{K}_{min}(T_1, T_2), k'\}$.

If $\mathcal{K}_{min}(T_1, T_2) \geq k'$, $j = \mathcal{K}_{min}(T_1, T_2)$. By Lemma 5, $\mathcal{C}_2(T_1, T_2, k) > 0$.
If $\mathcal{K}_{min}(T_1, T_2) < k'$, $j = k'$, thus $\mathcal{C}_2(T_1', T_2, j) = \mathcal{C}_2(T_1', T_2, k') > 0$:

$$\mathcal{C}_2(T_1', T_2, j) = \frac{T_1' V_1(R_j)}{T_2 V_2(R_j)} - \frac{v_{1,j+1} \cdot V_2(L_j)}{v_{2,j+1} \cdot V_1(L_j)} > 0$$

Remember $T_1' < T_1$, so:

$$\begin{aligned} \mathcal{C}_2(T_1, T_2, j) &= \frac{T_1 V_1(R_j)}{T_2 V_2(R_j)} - \frac{v_{1,j+1} \cdot V_2(L_j)}{v_{2,j+1} \cdot V_1(L_j)} \\ &> \frac{T_1' V_1(R_j)}{T_2 V_2(R_j)} - \frac{v_{1,j+1} \cdot V_2(L_j)}{v_{2,j+1} \cdot V_1(L_j)} \\ &> 0 \end{aligned}$$

So we have proved that $\mathcal{C}_2(T_1, T_2, j) > 0$ for $j = \max\{\mathcal{K}_{min}(T_1, T_2), k'\}$. If $\mathcal{C}_1(T_1, T_2, j) > 0$, then Statement 1 is true, otherwise Statement 2 is true, i.e., at least one of Statement 1 and 2 must be true.

**Inductive Step** Assume this lemma holds for $j \in [\max\{\mathcal{K}_{min}(T_1, T_2), k'\}, \mathcal{K}_{max}(T_1, T_2)]$, then we prove it also holds for $j + 1$. Assuming Statement 1 does not hold for $j + 1$, we show that $\mathcal{C}_1(T_1, T_2, j + 1) \leq 0$ and $\mathcal{C}_2(T_1, T_2, j + 1) > 0$, thus Statement 2 must hold for $j + 1$.

Assume Statement 1 does not hold for $j + 1$, which means there does not exist $k \in [\max\{\mathcal{K}_{min}(T_1, T_2), k'\}, j + 1]$, such that $\mathcal{C}_1(T_1, T_2, k) > 0$ and $\mathcal{C}_2(T_1, T_2, k) > 0$. Because $j \in [\max\{\mathcal{K}_{min}(T_1, T_2), k'\}, j + 1]$, then it must be $\mathcal{C}_1(T_1, T_2, j) \leq 0$ and $\mathcal{C}_2(T_1, T_2, j) \leq 0$. By the inductive assumption that the lemma holds for $j$, one of Statement 1 and 2 must hold, and we know that Statement 1 does not hold, so Statement 2 must hold for $j$, i.e., $\mathcal{C}_1(T_1, T_2, j) \leq 0$ and $\mathcal{C}_2(T_1, T_2, j) > 0$. Thus,

$$\mathcal{C}_1(T_1, T_2, j) \leq 0$$

$$\frac{v_{1,j} \cdot V_2(R_{j+1})}{v_{2,j} \cdot V_1(R_{j+1})} - \frac{T_1 V_1(L_{j+1})}{T_2 V_2(L_{j+1})} \leq 0$$

$$\frac{v_{1,j} \cdot V_2(R_{j+1})}{v_{2,j} \cdot V_1(R_{j+1})} \leq \frac{T_1 V_1(L_{j+1})}{T_2 V_2(L_{j+1})}$$

$$\frac{v_{1,j} \cdot V_2(L_{j+1})}{v_{2,j} \cdot V_1(L_{j+1})} \leq \frac{T_1 V_1(R_{j+1})}{T_2 V_2(R_{j+1})}$$

Remember that $\frac{v_{1,j+2}}{v_{2,j+2}} < \frac{v_{1,j}}{v_{2,j}}$, so:

$$\begin{aligned}
\mathcal{C}_2(T_1, T_2, j+1) &= \frac{T_1 V_1(R_{j+1})}{T_2 V_2(R_{j+1})} - \frac{v_{1,j+2} \cdot V_2(L_{j+1})}{v_{2,j+2} \cdot V_1(L_{j+1})} \\
&> \frac{T_1 V_1(R_{j+1})}{T_2 V_2(R_{j+1})} - \frac{v_{1,j} \cdot V_2(L_{j+1})}{v_{2,j} \cdot V_1(L_{j+1})} \\
&\geq 0
\end{aligned}$$

Remember we assume Statement 1 does not hold for $j + 1$, thus at most one of $\mathcal{C}_1(T_1, T_2, j+1) > 0$ and $\mathcal{C}_2(T_1, T_2, j+1) > 0$ is true. And we have shown $\mathcal{C}_2(T_1, T_2, j+1) > 0$, so it must be the case that $\mathcal{C}_2(T_1, T_2, j+1) > 0$ and $\mathcal{C}_1(T_1, T_2, j+1) \leq 0$, i.e., Statement 2 holds for $j+1$. $\square$

Finally, we are ready to finish the proof of Theorem 6. By Lemma 9, we know that one of Statement 1 and 2 must be true for $j = \mathcal{K}_{max}(T_1, T_2)$. By Lemma 4, Statement 2 is false for $j = \mathcal{K}_{max}(T_1, T_2)$, so Statement 1 must be true. Thus, we have proved that there exists an auto-bidder equilibrium with allocation $N_k$ with reported targets $(T_1, T_2)$, such that $k \geq k'$. $\square$

