# OpenReview forum: "Risk-Averse and Optimistic Advertiser Incentive Compatibility in Auto-bidding"
_ICML.cc/2026/Conference — ICML 2026 regular_

### Official Review · Reviewer_NLpU · 2026-03-09

**Soundness:** 3
**Presentation:** 2
**Significance:** 3
**Originality:** 3
**Overall Recommendation:** 4
**Confidence:** 4

**Summary:**

This paper introduces two refined notions of auto-bidding incentive compatibility—Risk-Averse Auto-bidding Incentive Compatibility (RAIC) and Optimistic Auto-bidding Incentive Compatibility (OAIC)—as relaxations of the original Auto-bidding Incentive Compatibility (AIC) concept proposed in prior work. The authors show that the Second-Price Auction (SPA) satisfies both RAIC and OAIC for multiple advertisers under both uniform and non-uniform bidding strategies. Additionally, they establish that the First-Price Auction (FPA) satisfies OAIC when advertisers employ non-uniform bidding strategies. The work contributes meaningfully to our understanding of mechanisms that incentivize truthful reporting of values by advertisers who are value maximizers subject to target cost-per-acquisition (tCPA) constraints—a critical issue as auto-bidding systems become increasingly prevalent in online advertising. It is particularly noteworthy that analyzing uniform bidding turns out to be significantly more technically challenging than non-uniform bidding.

**Compliance With Llm Reviewing Policy:**

Affirmed.

**Final Justification:**

The rebuttal addressed the specific technical ambiguities I pointed out. I will keep the weak accept recommendation.

**Key Questions For Authors:**

Q1. In the definition of incentive compatibility, the focus appears to be on deviations to larger tCPA constraints. Is there a straightforward argument showing that an advertiser would never benefit from reporting a smaller tCPA? If so, could the authors clarify this intuition?

Q2. Around lines 168–169, the claim that non-uniform bidding cannot improve value while meeting the ROI constraint is not obvious. Could the authors elaborate on why an advertiser cannot, for example, bid slightly lower in one slot and higher in another to achieve the same overall ROI while increasing total value?

Q3. For FPA with non-uniform bidding, the paper only establishes OAIC. Given that the proofs for RAIC and OAIC appear symmetric in earlier theorems (e.g., for SPA), why does this symmetry break down for FPA? Does FPA fail to satisfy RAIC in this setting, or was it simply not analyzed?

**Limitations:**

L1. The analysis omits two practically relevant scenarios: (i) FPA under uniform bidding, and (ii) a complete characterization of RAIC/OAIC for SPA with multiple bidders using uniform strategies. A counterexample or proof for these cases would strengthen the paper’s applicability.
L2. The main theoretical results lack an intuitive proof sketch or structural overview, making them harder to follow.

**Strengths And Weaknesses:**

Strengths
S1. The paper introduces two well-motivated relaxations of auto-bidding incentive compatibility—RAIC and OAIC—that capture realistic advertiser behaviors under uncertainty (e.g., risk aversion or optimism). These notions are theoretically grounded and meaningfully extend the original AIC framework.
S2. It provides rigorous theoretical analyses for both second-price and first-price auctions under distinct bidding strategies. Notably, the result that SPA satisfies both RAIC and OAIC when advertisers adopt uniform bidding strategy is non-trivial. The paper develops a novel analytical tool that characterizes the existence of equilibria via an equivalent condition for when one bidder wins a specific number of queries. This technical contribution provides a foundation for reasoning about equilibrium allocations in auto-bidding games and may be of independent interest to the mechanism design community.
S3. The problem addressed is highly relevant: as auto-bidding with tCPA constraints dominates modern ad platforms, understanding which auction formats align with truthful value reporting is crucial for mechanism design.

Weaknesses
W1. The paper does not analyze FPA under uniform bidding, nor does it fully address whether SPA satisfies RAIC/OAIC in multi-bidder settings with uniform strategies (e.g., via a proof or counterexample).
W2. Several parts of the proofs—particularly the main theorem—would benefit from a clearer high-level roadmap to improve readability.

---

> ### Author Rebuttal · Authors · 2026-03-26
>
> Thanks for the review!
>
> Q1: To clarify, we actually focus on deviations to *smaller* tCPA constraints (we checked the definitions and this seems to be the case; please let us know if that is not the case). We assume that advertisers have a hard tCPA constraint in mind and that violating it is a no-go. Therefore, they have no incentive to raise their tCPA constraint above their true target.
>
> Q2: To clarify, around lines 168 – 169, we are *not* saying that non-uniform bidding does not help (it can). What we are saying is that there is no reason for a bidder to bid *below* their value for any query. This is a standard argument which we will add but here is the argument. Basically, if the bidder (say bidder $i$) was bidding $< T_i \cdot v_{i,j}$ on query $j$ and wins then they can raise their bid to $T_i \cdot v_{i,j}$ and still win without changing allocation or payment. On the other hand, if they do not win then increasing their bid to at most $T_i \cdot v_{i,j}$ either does not change their allocation or strictly provides them with more value while preserving the constraints satisfied. So bidding at least $T_i \cdot v_{i,j}$ is good for the bidder. However, bidding exactly this on one query and more on another query might be good.
>
> Q3: To be honest, proving RAIC for FPA has turned out to be notoriously difficult. For OAIC, we can write an optimization problem that characterizes the best-case equilibrium. When the target is lowered, this only makes the optimization more constrained so the best-case equilibrium can go down. For RAIC, it is a lot more difficult and our intuition stems from trying to make the SPA RAIC proof work for FPA as well. The technical challenge is that the SPA RAIC proof starts from a bad equilibrium for advertiser 1 and constructs a worse equilibrium for advertiser 1 with a lowered tCPA. In FPA, this causes a chain reaction where other bidders best respond which in turn causes advertiser 1 to best respond, etc. For example, we haven’t been able to rule out that bidder 1 could lower its target to cause another bidder to take a query that bidder 1 does not like, causing it, in turn, to give up another query that bidder 1 does like, and therefore bidder 1 actually is better off.

---

> > ### Author Rebuttal · Reviewer_NLpU · 2026-04-03
> >
> > Thank you for your detailed rebuttal and the clarifications provided regarding my previous concerns.
> > I have carefully reviewed your responses to Q1, Q2, and Q3. Your explanations have successfully addressed the specific technical ambiguities I raised:
> > • Q1 & Q2: The clarification regarding the direction of deviations (towards smaller CPA constraints) and the logical justification for why bidders have no incentive to bid below $T_i \cdot v_{ij}$ is convincing. I appreciate your commitment to incorporating this standard argument more explicitly into the final manuscript.
> > • Q3: I acknowledge the significant technical challenges you outlined in proving Robustness against Advertiser Incentive Compatibility (RAIC) for First-Price Auctions (FPA). Hopefully you can resolve it in future work.
> > Overall, I will retain my score. The core contribution is sound and addresses a problem of practical significance and I believe the paper still requires refinement in its presentation before publication.

---

### Official Review · Reviewer_RASg · 2026-03-09

**Soundness:** 2
**Presentation:** 3
**Significance:** 2
**Originality:** 2
**Overall Recommendation:** 3
**Confidence:** 4

**Summary:**

This paper addresses a real and well-motivated problem in the auto-bidding literature: the original AIC definition of Alimohammadi et al. (2023) is too stringent, comparing worst-case truthful equilibria against best-case deviations, and consequently yields entirely negative results for standard mechanisms. The authors propose two natural relaxations — RAIC and OAIC — and show that SPA satisfies both under tCPA constraints.

**Compliance With Llm Reviewing Policy:**

Affirmed.

**Key Questions For Authors:**

Question 1. Why should we believe tCPA results tell us anything about practice when budget constraints — the most common type — break your entire technical apparatus (the undominated bids assumption fails)? Is there a fundamental reason budgets are hard here, or just a technical gap?

Question 2. RAIC and OAIC compare like-for-like equilibria (worst-vs-worst, best-vs-best). What happens when the advertiser is neither fully pessimistic nor fully optimistic — say, they believe equilibria are selected according to some distribution? Do your results survive any notion of "mixed" equilibrium selection, or do they depend critically on the extremes?

Question 3. Your main uniform-bidding results stop at two advertisers. Is the barrier combinatorial (the prefix-allocation structure breaks with three or more bidders) or algebraic (the mediant inequalities no longer suffice)? Concretely: do you have a counterexample showing SPA fails RAIC or OAIC with three uniform bidders, or do you conjecture the results extend?

**Limitations:**

The conceptual contribution (two natural relaxations of a prior definition) is modest, the model is restricted in ways that limit practical relevance (tCPA only, two bidders, single slot), and the technical results, while clean, do not reach the level of depth or surprise that I would expect for ICML.

The paper reads as a solid incremental contribution to the auto-bidding theory literature rather than a paper that changes how we think about the problem. I would encourage the authors to: (a) extend the results to budget constraints or explain the fundamental barriers; (b) connect the definitions to the broader implementation theory literature; (c) push beyond two bidders; and (d) develop the equilibrium characterization (Theorem 4) as a standalone contribution with broader applications. A substantially revised version addressing these points could be a strong paper.

**Strengths And Weaknesses:**

### Strengths

The paper identifies a genuine shortcoming in the prior literature. The AIC definition of Alimohammadi et al. (2023) compares the worst equilibrium under truth against the best under deviation — an asymmetry that is hard to justify from any coherent model of advertiser behavior. By pointing this out and proposing alternatives and proved that SPA works well for them, the paper makes a useful conceptual contribution to a growing and practically important literature.

Clean characterization of equilibria under uniform bidding (Theorem 4). The prefix allocation structure of equilibria is elegant and potentially useful beyond this paper. The mediant-based lemmas (Lemmas 1–3) that support Theorem 4 are clean and well-executed.

The paper's topic is practically relevant in application. Auto-bidding is now the dominant paradigm in online advertising. The positive results for SPA are reassuring, if somewhat expected given SPA's generally nice incentive properties.

### Weakness:

The model is restricted: only tCPA constraints. The paper explicitly excludes budget constraints, which are arguably the most common and economically important constraint in practice. The authors acknowledge (Section 5) that the undominated bids assumption fails under budget constraints, so the entire technical apparatus breaks down. This is a serious limitation for a paper whose motivation rests on practical relevance.

The main positive results (Theorems 5 and 6) hold only for two uniform bidders in SPA. And the analysis is for single-slot auctions only. But auto bidding is more interesting in the more general setting that relax both these assumptions. Can the author discuss how the results extend, or whether the prefix-allocation structure survives in richer settings?

The technical depth is adequate but not exceptional. Given that SPA are well known to have strong incentive properties in classical settings, the results are not particularly surprising. The paper would be more compelling if it uncovered a separation — e.g., a mechanism that is OAIC but not RAIC, or a setting where a widely-used mechanism fails one of these conditions in a practically meaningful way. The result that FPA is OAIC but not shown to be RAIC with non-uniform bidding hints at this, but is not developed sufficiently.

The paper would benefit from analysis based on ordinal preference hence extended to welfare. It would be helpful if developed more carefully on ordinal languages and showing that the preferences are robust to transformations of the advertiser's utility.

---

> ### Author Rebuttal · Authors · 2026-03-26
>
> Thanks for your thoughtful questions.
>
> Question 1: Budget constraints are also a very interesting problem and that makes sense as a follow up (we will mention one easy observation below that we also discussed in the last paragraph of the paper). However, we will note that tCPA is now widely accepted as being very standard. For example, [1] mentions that “the prevalent adopted model for the behavior of auto-bidding agents is that of value maximization.” In addition, there is quite a large body of work now that focuses on tCPA constraint so regardless of which constraint is the most common, tCPA is definitely well-established. In addition, even with a budget constraint, only one of (i) the tCPA constraint or (ii) the budget constraint is likely to hold at equilibrium. Our results would apply to the budget constraint setting where the tCPA constraint is the binding constraint.
>
> Going back to budget constraints, here is a simple example that shows that, in some sense, OAIC and RAIC are somewhat degenerate. For OAIC, a bidder can always bid infinity and have everybody else bid 0. This is an equilibrium and is clearly the bidder’s favorite equilibrium. So the bidder is indifferent among the different budgets. For RAIC, the argument is reversed, the bidder can bid 0 whereas one competitor bids infinity. Again, for RAIC, the bidder would be indifferent among the different budgets. It would be an interesting followup to refine AIC more for budgets in light of this.
>
> [1] Balseiro, S., Deng, Y., Mao, J., Mirrokni, V., & Zuo, S. (2021). Robust auction design in the auto-bidding world. Advances in Neural Information Processing Systems, 34, 17777-17788.
>
> Question 2: First, note that the fact that SPA satisfies RAIC and OAIC means it also satisfies some less-extreme definitions as well. For example, any convex combination of RAIC and OAIC, as the definition of AIC, is automatically satisfied as well.
>
> Another possible alternative, which we think would be nice future work, is to analyze the autobidding dynamics to see, for every target, what is the average value obtained in equilibrium. For example, we can assume autobidders use no-regret learning algorithms. Note that there is some work on understanding these dynamics (e.g. [2]) so it would be interesting to adopt those models for this question.
>
> [2] Paes Leme, R., Piliouras, G., Schneider, J., Spendlove, K., & Zuo, S. (2024, July). Complex dynamics in autobidding systems. In Proceedings of the 25th ACM Conference on Economics and Computation (pp. 75-100).
>
> Question 3: The barrier right now is mainly technical and we don’t have a strong reason to believe RAIC / OAIC breaks for more than two bidders. The proof relies on the fact that we can sort queries using the ratio of the two bidders’ values thus allowing us to use the mediant inequality. Generalizing this beyond two bidders would be a very interesting technical contribution.

---

> > ### Author Rebuttal · Reviewer_RASg · 2026-04-03
> >
> > The author stressed in their answer by citing several papers that only considering tCPA suffice.
> >
> > The questions raise (2) and (3) are still not answered. I maintain my score.

---

### Official Review · Reviewer_pDfc · 2026-03-10

**Soundness:** 4
**Presentation:** 3
**Significance:** 3
**Originality:** 3
**Overall Recommendation:** 5
**Confidence:** 4

**Summary:**

Paper offers new concepts of risk-averse and optimistic auto-bidding incentive compatibility. They show that Second-Price Auction (SPA) is RAIC and OAIC whereas previous works had found that SPA is not auto-bidding incentive compatible when there are tCPA constraints. They also show that First-Price Auction is OAIC. These allow definitions of ordinal preference over varying reporting targets.

**Compliance With Llm Reviewing Policy:**

Affirmed.

**Final Justification:**

Authors have addressed most of my concerns; I am happy to maintain my positive score.

**Key Questions For Authors:**

1.	It is stated that transitivity is not satisfied when multiple equilibria exist. I think this is a crucial point for why this paper matters, and it could use slightly more exposition or formal description of this violation in the notation of the paper.
2.	I understand this is a theoretical paper first and foremost, but I would enjoy more discussion of practical application, which is now limited to the first paragraph.

**Limitations:**

Paper is missing an impact statement, but they acknowledge future research directions. This include analyzing situations when there are more than two bidders or when tCPA constraints are relaxed.

**Strengths And Weaknesses:**

The paper is theoretical and offers advancement from previous papers, which relied on a restrictive definition of auto-bidding incentive compatibility. This is mainly useful for future works that may be interested in showing the usefulness of a mechanism, especially in an ordinal preferences setting.

Soundness:
Because this is mainly a theoretical paper that revises a definition for a certain context, it primarily consists of theoretical results, proofs of which are provided in the appendix. All claims are well supported. Any assumptions are supported with references to the literature.

Presentation:
All theoretical results are clearly stated in the paper,  and proofs are provided in the appendix with clear steps. In section 1, it is said that FPA is both RAIC and OAIC, but RAIC analysis is stated as being non-trivial.

Significance:
The authors state very briefly that the previous AIC definition presented in Alimohammadi et al. (2023) is not compatible with with ordinal preferences because transitivity is violated when there are multiple equilibria. Paper also applies their definitions to two popular mechanisms: SPA and FPA. To my understanding, RAIC and OAIC are simply statements about mechanisms that differ in level of strength. This paper is marketed mainly as a theoretical tool that could be used in future work to establish usefulness of mechanisms.

---

> ### Author Rebuttal · Authors · 2026-03-26
>
> Thank you for the positive evaluation of the paper!
>
> Regarding: transitivity. We can update the paper to address this a little thoroughly. We mainly wanted to drive home that the prior definition in the prior work could not have a notion of ordinal preferences so having a non-trivial notion of transitivity is also impossible. By their definition of AIC, a mechanism is not AIC if there exists a scenario that the best equilibrium by reporting a non-truthful target T’ is strictly better than the worst case equilibrium by reporting the true target T. Now let us try to define advertiser ordinal preference with this definition of AIC and then we can test whether transitivity holds with this definition of preference. Suppose T and T’ end up with the same set of equilibria values, where the advertiser gets a total value of W1 or W2, and W1 > W2. We will have the conclusion that the advertiser strictly prefers T to T’, and strictly prefers T’ to T at the same time. That is why we say that transitivity is not satisfied. If transitivity were to hold, this would imply the advertiser strictly prefers T to itself, which is a logical contradiction. That is why a valid transitive preference relation cannot be satisfied.
>
>
> Regarding: practical application. We can expand on it. The point is that there was been a lot of work on understanding the use of existing auction formats for a setting with autobidders without much consideration to incentives (there *are* papers on designing *new* auctions that take incentives into account, however). A previous paper showed that SPA for autobidders may actually have strong incentive issues whereas our paper has the practical implication that SPA may be fine in terms of incentives. Thus, this paper is more about the observation that we may not need to revamp existing auction systems to account for incentives with autobidders.

---

> > ### Author Rebuttal · Reviewer_pDfc · 2026-04-02
> >
> > Thank you for the response. I will maintain my positive score.

---

### Official Review · Reviewer_MKqC · 2026-03-11

**Soundness:** 3
**Presentation:** 3
**Significance:** 3
**Originality:** 3
**Overall Recommendation:** 4
**Confidence:** 2

**Summary:**

The paper studies incentive compatibility in auto-bidding auctions with tCPA constraints. It proposes two relaxed notions, RAIC and OAIC, and proves that the Second-Price Auction satisfies both under non-uniform bidding and uniform bidding with two advertisers.

**Compliance With Llm Reviewing Policy:**

Affirmed.

**Final Justification:**

Authors have addressed most of my concerns; I am happy to maintain my positive score.

**Key Questions For Authors:**

The analysis of uniform bidding is restricted to the two-advertiser case. What prevents the extension of these results to three or more advertisers? Is this merely a technical limitation of the proof technique, or does SPA actually fail to satisfy RAIC/OAIC under uniform bidding when there are three or more advertisers?

**Limitations:**

Yes

**Strengths And Weaknesses:**

Strengths
1.The paper identifies an important limitation in the standard definition of AIC and proposes two refined notions, RAIC and OAIC, that capture risk-averse and optimistic comparisons. This conceptual refinement is novel and addresses the non-transitivity issue inherent in the original AIC definition.
2.The technical treatment of the uniform bidding case, particularly the equilibrium characterization via sorting queries by value ratios and deriving prefix allocations, provides useful structural insights into the equilibrium behavior of auto-bidders under CPA constraints.
Weaknesses
1.The analysis of uniform bidding is restricted to the two-advertiser case (Theorems 4–6), which significantly limits the scope of the results. The paper does not provide sufficient intuition for why the approach cannot be extended to three or more advertisers, or whether this is due to technical difficulty or a fundamental limitation of the framework.
2.Assumption 1 states that bidding Ti*vi,j “obviously weakly dominates” any lower bid, but this claim is not formally justified. Since this argument appears to rely on equilibrium properties of the mechanism, a more explicit justification is necessary.
3.The proof of Theorem 4 is only sketched and omits several key arguments. In particular, the claim that the feasibility inequalities depend only on advertisers’ reported targets and values for the queries requires further clarification, given that bid multipliers are endogenous and determined in equilibrium.
4.The paper focuses almost entirely on theoretical properties and does not discuss implementation considerations or the practical implications of RAIC/OAIC for real-world auto-bidding systems.

---

> ### Author Rebuttal · Authors · 2026-03-26
>
> Thank you for the detailed review!
>
> 1. Right now, the analysis for two advertisers is restricted to two bidders primarily because of technical difficulties and the limitation of our proof framework. The proof relies on the fact that we can sort queries using the ratio of the two bidders’ values thus allowing us to use the mediant inequality. Generalizing this beyond two bidders would be a very interesting technical contribution.
>
> 2. We added a footnote about this but we can clarify more. Basically, if the bidder was bidding $< T_i \cdot v_{i,j}$ and wins then they can raise their bid to $T_i \cdot v_{i,j}$ and still win without changing allocation or payment. On the other hand, if they do not win then increasing their bid to at most $T_i \cdot v_{i,j}$ either does not change their allocation or strictly provides them with more value while preserving the constraints satisfied.
> Just one terminology clarification: “obviously weakly dominates” is a term which implies that the bidder can raise their bid to $T_i \cdot v_{i,j}$ without any consequence but with potential upside.
>
> 3. The proof of Theorem 4 is given in the appendix.
>
> 4. Thanks for this comment. There was been a lot of work on understanding the use of existing auction formats for a setting with autobidders without much consideration to incentives (there *are* papers on designing *new* auctions that take incentives into account, however). A previous paper showed that SPA for autobidders may actually have strong incentive issues whereas our paper has the practical implication that SPA may be fine in terms of incentives. A key takeaway from our work is that the current auction format already reasonably aligns with incentives.

---

### Decision · Program_Chairs · 2026-04-30

**Decision:**

Accept (regular)

**Comment:**

The paper studies incentive compatibility in autobidding scenarios. Reviewers highlighted the introduction of RAIC and OAIC as a meaningful refinement of prior AIC notions in multi-equilibrium settings. While reviewers are mostly positive about the results, some concerns remain on the limitations of the setting studied which may limit the practical scope of the results. I will recommend weak accept.